# VIC-CropSyst-v2: A regional-scale modeling platform to simulate the nexus of climate, hydrology, cropping systems, and human decisions

Keyvan Malek [1], Claudio Stockle [1], Kiran Chinnayakanahalli [3], Roger Nelson [1], Mingliang Liu [2], Kirti Rajagopalan [2], Muhammad Barik [2], Jennifer Adam [2]

[1]Department of Biological Systems Engineering at Washington State University
[2]Department of Civil and Environmental Engineering at Washington State University
[3]Air Worldwide Corp.

*Correspondence to*: Jennifer Adam (jcadam@wsu.edu) and Keyvan Malek (keyvanmalek@gmail.com)

**Abstract**

Food supply is affected by a complex nexus of land, atmosphere, and human processes, including short- and long-term
stressors (e.g., drought and climate change, respectively). A simulation platform that captures these complex elements can be used to inform policy and best management practices to promote sustainable agriculture. We have developed a tightly-coupled framework using the macroscale Variable Infiltration Capacity (VIC) hydrologic model and the CropSyst agricultural model. A mechanistic irrigation module was also developed for inclusion in this framework. Because VIC-CropSyst combines two widely-used and mechanistic models (for crop growth phenology, growth, and management; and
macroscale hydrology), it can provide realistic and hydrologically-consistent simulations of water availability, crop water requirement for irrigation, and agricultural productivity for both irrigated and dryland systems. This allows VIC-CropSyst to provide managers and decision makers with reliable information on regional water stresses and their impacts on food production. Additionally, VIC-CropSyst is being used in conjunction with socio-economic models, river system models and atmospheric models to simulate feedback processes between regional water availability, agricultural water management
decisions, and land-atmospheric interactions. The performance of VIC-CropSyst was evaluated at both regional (over the U.S. Pacific Northwest) and point scales. Point-scale evaluation involved using two flux tower sites located in agricultural fields in the U.S. (Nebraska and Illinois). The agreement between recorded and simulated evapotranspiration (ET), applied irrigation water, soil moisture, leaf area index (LAI), and yield indicated that, although the model is intended to work at regional scales, it also captures field scale processes in agricultural areas.

**Keywords:** integrated platform, hydrologic model, agricultural model, adaptation strategies, irrigation management, regional scale, climate change

| Nomenclature | | | |
|---|---|---|---|
| $E_{si}$ | Evaporation from soil during irrigation | $A_p$ | Irrigated covered area |
| $E_c$ | Evaporation from intercepted water | $D$ | Droplet size |
| $E_d$ | Evaporation from irrigation droplets | $V_0$ | Initial velocity of droplets |
| $T$ | Transpiration | $MAD$ | Maximum allowable depletion |
| $ET_p$ | Potential evapotranspiration | $g$ | Acceleration of gravity |
| $A_w$ | Soil wetted area | $Y_0$ | Height of nozzle |
| $T_i$ | Time of irrigation | $Y$ | Canopy height |
| $D_p$ | Deep percolation | $S$ | Sorptivity coefficient |
| $K_s$ | Saturated hydraulic conductivity | $B_i$ | Runoff calibration parameter |
| $R_o$ | Runoff loss | $D_s$ | VIC base flow calibration parameter |
| $q$ | Emitter discharge | $Ds_{max}$ | VIC base flow calibration parameter |
| $\rho_b$ | Soil bulk density | $W_s$ | VIC base flow calibration parameter |
| $\Delta\theta$ | Change in the water content | $GDD$ | Growing degree days |
| $LAI$ | Leaf area index | $E_s$ | Evaporation from soil |
| $K_c$ | Crop coefficient | $ET_a$ | Actual evapotranspiration |

10 **1. Introduction**

Projected increases in food demand (Godfray et al., 2010) along with other stressors such as droughts and extreme heat events contribute to threats on global food supply (Wheeler and Braun, 2013). Despite existing research on food scarcity, there are still unanswered questions about the relationship between food supply and the nexus of water resources, agriculture and human decisions. For example, how expectations of future climatic conditions influence farmer behaviour such as

15 capital intensive switches in technology or cropping systems, is not well understood. Such scenarios require a simulation tool

that can capture large-scale hydrologic processes while accurately simulating the impacts of climate, management, and water availability on different crop types. Moreover, regional consequences of decisions intended to mitigate the damages of future stressors are not well understood (Robertson and Swinton, 2005). For example, improvement in the efficiency of irrigation systems may increase consumptive water uses and lead to a reduction in return flow from irrigated areas (Causapé et al., 2004; Gosain et al., 2005). Return flow plays a significant role in the water availability of many agricultural regions; e.g., 40% of the water availability at the Yakima River's Parker Gauge in an average year is generated through return flows from upstream lands (USBR, 2010). Ecosystems and hydroelectric generation are also impacted as return flow changes. These knowledge gaps limit our ability to explore viable adaptation strategies, particularly in understanding unintended consequences. Integrated modeling platforms can contribute to the systems-level understanding of dynamics between agricultural processes, large-scale water resources management decisions, and land-atmospheric interactions.

The overall goal of this study is to develop a computational modeling platform that mechanistically captures the interactions between hydrology, crop growth and phenology, and crop and water resource management decisions in the context of global change. Such a platform allows for investigation around multiple objectives: 1) understanding how climate dynamics and land-atmosphere interactions affect water and agricultural sustainability; and (conversely) 2) exploring the role of agricultural (biophysical and socioeconomic) processes in driving land-atmosphere interactions, including climate feedback mechanisms at larger scales.

### 1.1. Future Food Demand and Supply

While over 800 million people throughout the world suffer from undernourishment (FAO, 2013), global change is expected to exacerbate food security problems. The demand for food is increasing due to population growth and changes in food dietary tendency towards higher consumption of meat products (Long et al., 2015). Food supply, on the other hand, may not increase as fast as demand (Wheeler and Braun, 2013), as it is affected by complicated interactions between climate, the hydrologic cycle, cropping systems, and human decisions. Table 1 shows the variety of ways that climate change can impact crop yield, with some impacts being positive and others negative; the net result is dependent on region, crop, and future time period. Mechanistic integrated modeling platforms are necessary to assess the net impact of global change on crop production.

[TABLE 1]

### 1.2. Interactions between Cropping Systems, the Hydrologic Cycle, Climate, and Human Decisions

Although agricultural productivity is affected by disturbances in the regional cycles of water and energy (Pielke Sr. et al., 2007), agriculture itself feeds back to alter the hydrological cycle by changing evapotranspiration (ET), and the magnitude and temporal regime of soil moisture, infiltration and runoff generation (Haddeland et al., 2006; Harding et al., 2015; Lu et al., 2015; Sorooshian et al., 2012). The impact of irrigated agriculture on energy and water cycles is particularly important

(Ferguson and Maxwell, 2011; Lobell et al., 2009; Pokhrel et al., 2016; Puma and Cook, 2010; Scanlon et al., 2007; Sridhar, 2013). Irrigation uses 70% of total global water withdrawals (Rost et al., 2008) and boosts soil moisture storage available for crop uptake, and ultimately increases ET. Irrigation losses also increase the amount of deep percolation and runoff.

While farmers can adjust their management decisions to reduce the negative impacts of climate change (e.g., switching to more-efficient irrigation technologies, more drought-tolerant crop types, varieties with longer growing periods, and precision agriculture), these human decisions can result in unintended impacts on regional water and energy cycles. The consequences of anthropogenic disturbances (e.g., irrigation withdrawal and dam construction) on the regional water cycle can be greater than the impacts of climate change (Haddeland et al., 2014). Irrigation management and changes in cropping patterns are two examples of management decisions influencing the amount of evapotranspiration, runoff, deep percolation, and soil moisture, all of which can alter timing and magnitude of return flow. In many agricultural basins, the availability of water for downstream users depends greatly on the return flow from upstream lands, which mainly comes from non-evaporative, reusable loss of water through conveyance systems and field-level application of irrigation water. Therefore, regional-scale simulation of the hydrologic cycle is crucial to the analysis of the impacts of water management in large river basins with significant agricultural activities.

VIC-CropSyst provides an advantage over the stand-alone CropSyst model when run over larger scales. Here, we define large-scale results as regionally-aggregated responses of agriculture to changes that can impact scales greater than a single cultivated field, such as a policy change (e.g., water law), climate-related impacts (e.g. warming-induced reductions in summer water availability), or development of large-scale infrastructure (e.g., a large reservoir). Allen et al., (2015) interviewed around twenty stakeholders, including governmental and non-governmental agency staff and producers, to understand their priorities, concerns and decision-making processes. They found that many of these stakeholders, including individual producers, are interested in local and basin-scale information about the impacts of climate change, infrastructural developments, and land management practices on the quantity, quality and temporal regimes of water resources. Therefore, large-scale integrated modeling platforms are also needed to inform regional natural and agricultural resource management policies and actions.

### 1.3. Agricultural Processes within Macroscale Hydrologic Models

#### 1.3.1. Capturing Cropping Systems within Land Surface Models

Land Surface Models (LSMs) are used for regional to global-scale simulations of water and energy cycles, often providing terrestrial boundary conditions to general circulation models (GCMs). Results of modeling studies have indicated that, despite the tremendous advances in earth system modeling, the current state of LSMs is not capable of capturing agricultural processes in a detailed manner (e.g. Chang et al., 2014; Haddeland et al., 2006; Hansen et al., 2006; Lobell et al., 2009, 2008; Ozdogan et al., 2010a). In many of them, agricultural processes are similar to natural vegetation (Chang et al., 2014); due to phenological similarities, agricultural lands are often represented by grass vegetation (Elliott et al., 2014). Also,

management or harvesting activities as well as $CO_2$ fertilization effects may be ignored (Drewniak et al., 2013). Mitchell et al. (2004) compared the results of four different models and reported poor overall performance among LSMs in capturing warm season ET. In most cases, this inconsistency can be explained by weak representation of agricultural processes. For example, Schwalm et al. (2010) compared 22 terrestrial biosphere models with North American flux tower sites and found

the performance of models in natural vegetation areas to be better than in cropland areas.

Bierkens (2015) reviewed twenty three global/large-scale hydrological models (GHMs; e.g. WaterGAP, Verzano et al., 2012; WBMPlus, Wisser et al., 2010; Mac-PDM.09, Gosling and Arnell, 2011; and H08, Hanasaki et al., 2010), LSMs (VIC, Liang et al., 1994;  MATSIRO, Takata et al., 2003; LM3, Milly et al., 2014; NOAH, Liu et al., 2016; JULES, Best et al.,

2011; CLM, Fisher et al., 2015; SiB, Baker et al., 2008; and ORCHIDEE, Vérant et al., 2004) and dynamic vegetation models (DVMs;  e.g. LPJmL, Fader et al., 2015). Among these models H08, MATSIRO, JULES, ORCHIDEE, and SiB use simple crop growth modules to simulate natural vegetation or generic C3 and/or C4 crops.  NOAH, CLM, and LPJmL have more sophisticated crop growth schemes; these are further discussed below.

Using prescribed seasonally and spatially variable leaf area index (LAI) and root density, Wei et al. (2013) modified

aerodynamic and soil deficit thresholds in the NOAH land surface model, thereby improving the simulation of warm season processes. In their model, however, crop growth and development does not mechanistically respond to climate, $CO_2$ concentrations, and soil moisture; this limits the accuracy of model simulation over agricultural areas where the feedback between agricultural processes and hydro-climatic conditions is significant.  Liu et al. (2016) improved simulation of crop processes in the NOAH-MP-Crop model but their model could only simulate corn and soybean and did not capture irrigation

processes.

Drewniak et al. (2013) enhanced the Community Land Model (CLM) in agricultural areas by using an improved representation of crop processes, but $CO_2$ fertilization effects, irrigation, and other common management activities were neglected. In their simulations, they considered only three crop species (wheat, corn and soybean) and used a fixed planting date, which can lead to a discrepancy with observations in that actual planting dates vary in time as a function of weather

(Zeng et al., 2013), and can result in an over-estimation of the negative impacts of warming on crop yield, as an earlier planting date is a viable adaptation strategy in many regions of the world (Waha et al., 2013). While a newer version of CLM (CLM4-Crop; Lu et al., 2015) simulates irrigation events and $CO_2$ fertilization as well as biomass and vegetation growth processes, its application is also limited to three crop types (Chen et al., 2015) and is not able to mechanistically simulate irrigation efficiency.

Elliott et al. (2014) compared ten GHMs and six global gridded crop models (GGCMs); they reported that the performance of GHMs is generally poor in the simulation of future irrigation water demand. Many of them use prescribed crop growth parameters and did not capture $CO_2$ fertilization nor sensitivity to heat and water stresses; the only exception was the Lund-Potsdam-Jena Managed Land Dynamic Global Vegetation and Water Balance Model (LPJmL), which is a hydrologic model that can mechanistically simulates both hydrologic and agricultural processes.  However, LPJmL simulates a limited number

of crops (Elliott et al., 2014) and, as compared to specialized crop models (e.g. CropSyst; the Decision Support System for Agrotechnology Transfer; DSSAT, Jones et al., 2003; and the Environmental Policy Integrated Climate Model; EPIC, Williams et al., 1989, 1983), uses more simplistic methods to simulate crop processes such as LAI development, root distribution, and the number of stressors considered (Rosenzweig et al., 2014; Stöckle et al., 2003). Moreover, although LPJmL is a grid-based model, so far it has been used to address global scale issues at coarse scale (0.5 °), and has not been tested and used for regional studies. Also LPJmL uses prescribed country-specific irrigation efficiency (Biemans et al., 2011; Fader et al., 2015; Rosenzweig et al., 2014), which can cause biases when LPJmL is applied at finer spatial scales. It is also worth mentioning that the scientific community has already benefited from watershed-scale hydrologic-agricultural models. For example the Soil Water Assessment Tool (SWAT Arnold et al., 1998; Neitsch et al., 2011) is coupled to a simplified version of the EPIC model (Williams et al., 1989, 1983) and is able to capture agricultural processes and management decisions. SWAT's shortcoming is the fact that it has seven crop classes and does not differentiate among crops within a class (e.g., tree fruits). Furthermore, SWAT uses predefined irrigation losses and does not simulate irrigation processes mechanistically.

### 1.3.2. Capturing Irrigation Systems within Land Surface Models

Irrigation is one of the important but under-appreciated processes in LSMs (Gordon et al., 2008; Ozdogan et al., 2010; Pokhrel et al., 2016). Normally, irrigation processes are treated in LSMs with one of the followings approaches. *1- Irrigation time and amount is not mechanistically simulated:* In most modeling studies, irrigation requirements are calculated using published irrigation guidelines or a time series of satellite observations (Pokhrel et al., 2011). In other models, irrigation water scarcity is not captured (e.g. Ozdogan et al., 2010), which can result in less realistic irrigation management during droughts. *2- Irrigation is included but with unrealistic assumptions of irrigation efficiency:* For example, CLM v4 simulates the time of irrigation based on soil deficit but does not consider irrigation losses (Leng et al., 2013). This can cause poor representation of hydrologic processes in agricultural areas and underestimation of irrigation demand. *3- Partitioning of overall efficiency into different losses through prescribed ratios:* Pokhrel et al. (2011) developed an irrigation module and coupled it to the Minimal Advanced Treatments of Surface Interaction and RunOff (MATSIRO) model. The irrigation module considers soil moisture deficit to calculate the time of irrigation, but their irrigation module did not consider the partitioning of the overall efficiency into different losses and did not simulate the dynamics between irrigation losses and the hydrologic cycle. Haddeland et al. (2006) implemented a simple irrigation module into the VIC model. This irrigation module, however, was limited to prescribed losses of sprinkler systems. Also, because the stand-alone VIC model does not mechanistically simulate crop processes, the timing and amount of the irrigation water is not responsive to crop growth, management, and phenology.

These shortcomings, simplifying assumptions, and lack of a mechanistic way to simulate irrigation processes in LSMs lead to inaccurate ET and water demand simulations (Pokhrel et al., 2011; Sridhar, 2013). Also, because LSMs are often coupled

to atmospheric models, this lack of capturing mechanistic irrigation processes will cause biases in turbulent heat flux simulations, leading to GCM errors.

## 2. Approach

Here, we introduce the newly integrated model VIC-CropSyst, which is a coupling between the VIC hydrologic model and the CropSyst crop growth, phenology, and management model. VIC-CropSyst can be used for regional to global-scale simulations of water and energy cycles over natural and managed terrestrial ecosystems. A process-based irrigation module was also developed to simulate the interactions between irrigation management decisions and the hydrologic cycle in this integrated model.

### 2.1. Descriptions of Stand Alone Models

#### 2.1.1. VIC

The VIC model is a processed-based large-scale hydrologic model developed initially by Liang et al. (1994). VIC uses the variable infiltration capacity curve introduced by Zhao et al. (1980) to simulate infiltration and surface runoff, and Franchini and Pacciani's (1991) formula to calculate base flow. Liang et al. (1996) further developed the model to represent multiple soil moisture layers (the original version only had two). Cherkauer et al. (2003) added processes for more accurate simulation of soil freeze and thaw as well as the canopy energy balance in freezing conditions; further information on simulation of the snowpack can be found in Andreadis et al. (2009). While simulation time-step of the stand alone VIC model can be specified to be daily, hourly or sub-daily (e.g. three hour), in the version of VIC-CropSyst described herein, the simulation time-step is currently limited to daily time-steps. Subsequent VIC-CropSyst model developments will allow for sub-daily time-steps. VIC also has the flexibility to be implemented over multiple resolutions (generally at or greater than $1/16^{th}$ °), and captures sub-grid heterogeneity in vegetation, elevation, snow depth, and a variety of other variables. The stand-alone VIC model uses prescribed monthly LAI values to represent seasonal variations of vegetation cover, and so does not simulate agricultural processes such as crop development and biomass production and the impacts of water, heat and nutrient stresses on crop growth. Also, the VIC model does not mechanistically simulate irrigation losses and only includes one type of irrigation (sprinkler). This limits VIC's ability to accurately simulate water demand, transpiration and agricultural productivity. VIC has been applied and evaluated by several researchers over a variety of areas; e.g., Elsner et al., (2010) and Hamlet and Lettenmaier (1999) over the Columbia River basin; Adam et al. (2007) in the Eurasian arctic; Maurer et al. (2002) over the contiguous U.S.; and Yuan et al. (2004) over China.

#### 2.1.2. CropSyst

CropSyst (Stockle et al., 1994, Stockle et al., 2003) is a process-based cropping system model, capturing water, nitrogen and carbon cycles as well as the key processes related to crop phenology, root and shoot growth, and biomass production and

yield. CropSyst simulates field operations including irrigation, fertilization, tillage, residue management and crop rotation. It also captures the effects of $CO_2$ concentration and stressors such as water limitation, temperature extremes and soil salinity on crop development. CropSyst has been applied over a range of climatic conditions worldwide, as well as for climate change studies (e.g., Confalonieri and Bocchi (2005) for rice in Italy; Ferrer et al. (2000) for corn in Spain; Pala et al. (1996) for wheat in Syria; Karimi et al., (2017) and Pannkuk et al., (1998) for wheat in the U.S. Pacific Northwest; and Alva et al. (2010) for potatoes in the U.S. Pacific Northwest). In CropSyst the daily biomass production is restricted to the minimum of the two following biomass generation routines: i) radiation-based biomass production, and ii) transpiration-based biomass production. After simulation of potential biomass, CropSyst takes water, heat, freezing and nutrient stresses into account to calculate the actual yield. These stresses also modify other crop processes such as transpiration and LAI. Stress sensitivity varies during different phonological periods (e.g. from flowering to maturity). Root occurrence varies in each of the soil layers and depends on the root growth deeper into the soil during biomass development; thus, crop water and nutrient uptake also varies by soil layer. While the start and last date of the growing period is an input to the model, actual crop growth starts after a certain amount of thermal accumulation has been achieved during this user-specified growing period. Crop growth and development is also a function of thermal accumulation, affecting actual harvest date and other growth stages.

### 2.2. Model Integration

We coupled the VIC version 4.1.2-e with CropSyst-v4.15, although the coupled model will be updated with new versions of VIC and CropSyst as they become available. In a spatially-explicit manner, VIC-CropSyst is able to capture a large variety of crop groups: 1- cereal grains (e.g. winter and spring wheat, corn, barley, oats, sorghum), 2- vegetables and melons (e.g. dill, radish, mint, broccoli, cauliflower, cabbage, carrot, onion, cucumber and pumpkins, watermelon), 3- fruits and nuts (e.g. plum, apricot, cherry, grape, walnut, pear, peaches, apples, blubbery, strawberry, cranberry), 4- root crops (e.g. potato, sugar beet), 5- leguminous crops (e.g. green and dry bean, lentil, chickpea, pea), 6- forages (e.g. pasture, alfalfa, hay, grass, clover, grass), and 7- oil seeds (e.g. soybean, mustard, sunflower). In the tightly coupled VIC-CropSyst model (Figure 1), all hydrologic processes except for transpiration are handled by VIC, while crop growth, transpiration, phenology, and management are handled by CropSyst. In the following section we briefly explain the structure of the VIC-CropSyst coupling (Figure 1). Then we discuss some of the changes we have made to each model to support this integration. Finally, we discuss the irrigation module that we have developed and implemented in VIC-CropSyst.

[FIGURE 1]

### 2.2.1. Water and Energy Balances in VIC-CropSyst

Figure 2 shows how VIC-CropSyst handles the water and energy budgets. VIC first simulates the energy balance (explained by Cherkauer et al., (2003) and Liang et al., (1994)). It estimates available energy per time step and uses an iterative

approach to partition the available energy into each of the energy components (e.g., snowmelt and sublimation heat fluxes, ground heat flux, and sensible heat flux). After these terms are calculated, the remaining energy will be available to potential evapotranspiration (ETp). Evaporation can happen from at least one of the five following processes (Thompson et al., 1993): 1- directly from irrigation water ($E_d$), 2- from intercepted water by the canopy ($E_c$), 3- from the wetted soil surface during

irrigation ($E_{si}$), 4- from the soil surface when irrigation is not occurring ($E_s$), and 5- transpiration ($T$).

CropSyst is called while VIC is simulating the energy balance, but after ETp is portioned into each of its terms. Following this, potential transpiration and availability of soil moisture are passed to CropSyst (Figure 2). Actual transpiration depends on the availability of soil water. When the soil does not have enough water to meet crop demand, actual crop transpiration is

less than potential. In the coupled model, CropSyst simulates actual transpiration, soil water extraction from each layer, water stress, and crop growth; it then passes the extracted soil water amount to VIC to calculate the water balance. VIC updates soil moisture and simulates the rest of the hydrologic components such as runoff and baseflow.

[FIGURE 2]

### 2.2.2. Significant Changes to Each Model

*Soil Hydrology:* In the integrated VIC-CropSyst model, CropSyst's soil hydrology is turned off, allowing VIC to simulate soil hydrologic processes, including the movement of water in soil, bare soil evaporation, and the generation of runoff and baseflow. We did this to retain consistency in all of the hydrologic processes. Standalone VIC and CropSyst use different

soil hydrologic assumptions to simulate processes related to soil water movement and the generation of runoff and baseflow; these inconsistencies can lead to an inaccurate simulation of irrigation demand and crop productivity. Because crop processes are sensitive to soil moisture availability, we have modified the VIC soil structure. While VIC previously had the capacity to handle an indefinite number of soil moisture layers, the majority of VIC applications utilize three layers, where runoff and baseflow are generated from the top and bottom layers, respectively, while the middle layer is the root zone where

plant water uptake occurs. Because the availability of water where roots are concentrated is central to unstressed crop growth, and because the dynamic simulation of root growth is sensitive to the vertical distribution of soil moisture, VIC's conventional three layering system is too coarse to accurately represent this condition, particularly during droughts and over rain-fed cropland. Therefore, we expanded the middle layer of VIC to 15 layers. Finally, the minimum soil moisture in VIC-CropSyst is set to the wilting point (except in the top evaporative layer).

*Soil File:* The conventional versions of VIC directly read soil properties (e.g. soil hydraulic conductivity, field capacity, wilting point, bulk density) from input files. For a more consistent way (between VIC and CropSyst) of inputting soil input

information, empirical functions developed by Saxton et al. (1986) were implemented in the model and VIC-CropSyst internally estimates the necessary soil parameters using soil textural characteristics (i.e., sand and clay percentages).

## 2.3. Irrigation Module

The irrigation module (Figure 3) is briefly explained below, while a more detailed description can be found in Malek et al (2016, in prep). The irrigation module calculates irrigation frequency, amount, and losses.

[FIGURE 3]

Currently, VIC-CropSyst simulates four major categories of irrigation systems: surface, center pivot, sprinkler, and drip. Each category includes subcategories. Drip systems include surface and subsurface drip irrigation. In surface drip irrigation, water is applied on the soil surface, while in subsurface drip irrigation, water is applied below the surface and will not lead to any soil evaporative losses. Surface irrigation includes furrow, rill, and border irrigation, and the main difference between these three systems is in their wetted surface area, which is smaller in a furrow system. Center pivots are represented by

eighteen different types of sprinklers that fall into two subcategories: impact and spray sprinklers. Impact sprinklers generally have a greater discharge rate and wetted radius. Sprinkler systems in VIC-CropSyst include seventeen nozzles from three major subcategories: solid set, big gun, and moving wheels. The subcategories differ in terms of discharge, wetted diameter, height, droplet size, and other aspects. The characteristics of these systems have been collected from different scientific papers, reports, and commercial catalogs, including Nelson Co. (2014) and RainBird (2014). This level of detail

offers a more accurate representation of irrigation practices, and it will help users to simulate the adaptation of different irrigation and management scenarios.

### 2.3.1. Irrigation Frequency

Evaporation, transpiration, and deep percolation cause reductions in root-zone soil water content. When soil moisture deficit reaches one of the following two thresholds, VIC-CropSyst triggers an irrigation event: 1- capacity of the irrigation system, which sets the maximum amount of water that can be applied in an irrigation event, and 2- the Maximum Allowable Depletion (MAD), which determines what degree of soil dryness causes water stress in each crop. To define crop-specific

MADs, we created a table of parameters using FAO-56 (Allen, 1998).

### 2.3.2. Evaporative Losses

In the drip and surface categories, evaporative losses happen only from the soil surface because irrigation happens below the canopy level. Irrigation takes place above the canopy in sprinkler and center pivot systems; therefore, evaporation from canopy-intercepted water ($E_c$) and the direct loss from droplets ($E_d$) are considered as major irrigation losses. VIC-CropSyst neglects evaporative losses from soil ($E_{si}$) for sprinkler and center pivot systems because energy is more readily available for water above the canopy and it suppresses the below-canopy evaporation (Uddin et al., 2013; Yonts et al., 2000). Evaporative losses from drip and surface irrigation systems are based on the following formula,

$$E_{si} = ET_p\, A_w\, T_i/24 \tag{1}$$

where $ET_p$ is potential ET [mm/$\Delta t$]; $A_w$ is the wetted surface fraction during irrigation; and $T_i$ is the time of irrigation [hr]. While $A_w$ is assumed to be 1.0, 1.0 and 0.5 for border, basin and farrow irrigation, respectively, we used Malek and Peters (2011)'s equation to estimate the wetted radius of drip irrigation and calculate the wetted percentage.

The following formulas are used to calculate $E_c$ and $E_d$ from sprinkler and center pivot irrigation systems. *Evaporation from Irrigation Intercepted Water ($E_c$)*: To calculate Ec, VIC-CropSyst uses the original VIC method (Liang et al., 1994). avoid overestimation of Ec in agricultural areas, we used the equation developed by Kang et al. (2005) to set maximum Ec. *Evaporation from Irrigation Droplets ($E_d$):* Users have the option to calculate Ed using one of two methods of:

1- Malek et al. (in review)

$$E_d = ET_p \times \left(\frac{1}{D}\right)^{0.52} \times \left(\frac{V_0 \sin(\theta)}{g} + \frac{\sqrt{V_0^2 \sin^2(\theta) + 2g(Y_0 - Y)}}{g}\right)^{1.57} \tag{2}$$

where $Y_0$(m) is height of nozzle; Y (m) is canopy height; $V_0$ (m/sec) is initial velocity of the irrigation water which depends on irrigation system pressure H (m), nuzzle coefficient $c_d$, and initial angle of sprinkler $\theta$; $A_p$ is irrigated area at a time; D (mm) is the droplet diameter and $ET_p$ (mm/$\Delta t$) is potential evapotranspiration.

2- Playán et al., (2005):

For sprinkler: $\quad E_d = 20.3 + 0.214\, U^2 - 0.00229\, RH^2 \tag{3}$

For moving laterals and center pivot:

$$E_d = -2.1 + 1.91\, U^2 + 0.231\, T \tag{4}$$

where T (°C) is the air temperature; U ($m\, s^{-1}$) is wind speed; and RH (%) is the relative humidity.

### 2.3.3. Deep Percolation Loss (D_p)

Dp is defined as irrigated water which penetrates below the root zone. Therefore, after an irrigation event the amount of water that enters the base flow layer and becomes inaccessible for crop roots is considered a deep percolation loss.

### 2.3.4. Runoff Losses ($R_o$)

Ro depends on soil infiltration rate and irrigation intensity. Whenever irrigation intensity is higher than soil infiltration capacity, runoff is generated as follows,

$$R_o = \frac{Ir}{t_{irr}} - f \tag{5}$$

where $f$ is the infiltration rate $\left(\frac{mm}{hr}\right)$, $I_r$ is the amount of irrigation water applied in each event $(mm)$ and $t_{irr}$ is the duration of irrigation $(hr)$. Although irrigation intensity is usually a management decision, soil texture and hydraulic conductivity are assumed to be the key considerations in a well-managed irrigation system; therefore in the beginning of simulation, VIC-

CropSyst estimates the irrigation duration ($I_{du}$) using the soil characteristics of each grid cell. The calculated $I_{du}$ is used to estimate the infiltration opportunity time of surface irrigation, rotation time in center pivot, and overlap and layout of sprinklers in solid-set, wheel move and big-gun irrigation systems. If approximated irrigation intensity exceeds the irrigation infiltration rate ($f$), the extra water generates runoff. VIC-CropSyst uses the equation developed by Philip, (1957) to estimate the infiltration rate,

$$f = \frac{1}{2} S T_i^{-0.5} + K_s \tag{6}$$

where K_s $\left(\frac{mm}{hr}\right)$ is the hydraulic conductivity and S is the sorptivity which is estimated using the Rawls et al., (1992) formula and is calculated based on soil texture and initial water content. Therefore, in VIC-CropSyst, Ro depends on

irrigation system, soil type, initial soil moisture as well as the intensity of water reaching to soil. Details of the runoff calculations are presented by Malek, et al. (2016, in prep).

### 2.4. Deficit irrigation

VIC-CropSyst's deficit irrigation module requires two main inputs: a) a first approximation to the irrigation water demand

obtained by generating time series of irrigation under no water stress condition using VIC-CropSyst, and b) deficit fractions

that indicate the water availability. VIC-CropSyst then reads the amount of recorded irrigation from step one and applies the deficit fraction to simulate the agricultural and hydrologic processes under realistic water deficit conditions. The deficit fraction can be either homogenously applied across the entire basin or separately specified for each farmer depending on water rights or other considerations.  Also, VIC-CropSyst can apply the deficit fraction during different times of the year. For example, if the water deficit happens later in the season, VIC-CropSyst can adjust irrigation amounts according to the timing of water shortage.

VIC-CropSyst has also been used in conjunction with reservoir models (e.g. ColSim; Wittwer et al., 2001 and YAK-RW; Zagona et al., 2001) to calculate the deficit irrigation fraction (e.g. Barik et al., 2017; Malek, et al., in preparation; Rajagopalan et al., in preparation). In general, the following six steps can be used to calculate and apply a deficit fraction: 1) VIC-CropSyst simulates the hydrologic states such as runoff and base flow as well as the irrigation water demand, 2) a routing model (i.e. Lohmann et al., 1998) is used to simulate streamflow, 3) simulated flow is bias corrected against observed flow, 4) a river system model is used to include operation of dams and reservoir and estimate water availability, 5) the availability of water is compared with demand, and 6)a deficit fraction is calculated and VIC-CropSyst is run to simulate the impacts of irrigation deficit on the hydrologic cycle and crop yields.

### 2.5. Previous versions of VIC-CropSyst

VIC-CropSyst v1.0 was originally developed and used to forecast the impact of climate change on Columbia River Basin water supply and irrigation water demand  (Yorgey, et al., 2011; Rajagopalan, et al., in review). This version was created using VIC (v4.0.7) and CropSyst (v4.15). This version is a lower coupling in terms of hydrology; i.e., both models simulate their own soil moisture with different soil parameters and soil layers. While VIC provides the water and cropping information and available energy for evapotranspiration, partitioning of energy to different evaporative losses (i.e., evaporation from soil and transpiration) is separately done in each model and irrigation evaporative losses are not considered in VIC's energy balance. The irrigation efficiencies were hard coded in this earlier version. VIC-CropSyst v1.1 was slightly modified and used by (Liu et al., 2013). Rajagopalan et al. (in preparation) also used VIC-CropSyst v1.1 to evaluate the impact of climate change on agricultural productivity in the CRB.  This manuscript describes the fully coupled version of the VIC-CropSyst model (version 2). This version is tightly connected in which VIC handles all of the soil hydrologic processes; to do this, some VIC soil processes were altered to be more compatible with CropSyst. Furthermore, the influence of crop transpiration on energy balance is captured in this new version. Finally, this version mechanistically simulates irrigation processes and losses (e.g., irrigation evaporative losses) and is able to apply deficit irrigation.

### 2.6. Data and Study Sites/Areas

VIC-CropSyst's simulated soil moisture, ET, yield and irrigation water demand were compared to observed data obtained from the FLUXNET network (Baldocchi et al., 2001). Simulated LAI was evaluated against Moderate Resolution Imaging

Spectroradiometer (MODIS) remote sensing observations (Cohen et al., 2006). We also evaluated regional performance of VIC-CropSyst in simulation of ET over the U.S. Pacific Northwest, including the states of Washington, Idaho and Oregon. Other studies such as Malek et at (in preparation a and b), Rajagopalan, et al., (in preparation), Barik et al., (2017), Hall et al., 2017), Yorgey et al., (2011) evaluated VIC-CropSyst in its capability to capture regional irrigation demand, naturalized

streamflow, observed flow, county level yield, snow water equivalent, and  irrigation efficiency.

### 2.6.1. Site Description

The flux tower stations considered in this study are located in two U.S. states of Nebraska (NE) and Illinois (IL) (Figure 4). Available environmental and agricultural information include latent heat, soil moisture and meteorological data, crop type,

LAI, and biomass production. The towers are all in agricultural fields and have relatively long periods of available data. The station in the IL is not irrigated and the site in NE is irrigated with recorded irrigation frequency and amount.

### 2.6.2. Meteorological, Soil, Land Cover, and Topographic Data

Daily meteorological data were acquired from the DAYMET (Thornton et al., 2012) gridded data source. Soil files were

taken from Maurer et al. (2002) for associated grid cells. We replaced its sand content with data available at the study site. We also added the clay percentages to Maurer et al. (2002)'s soil file. In our simulation, VIC-CropSyst reads the sand and clay content and uses pedo-transfer functions developed by Saxton et al. (1986) to generate saturated hydraulic conductivity, bulk density, air entry potential, the b coefficient of Campbell, (1974)'s soil retention curve, field capacity, wilting point, and porosity. Table 2 shows soil texture calculated using the United States Department of Agriculture's soil triangle (Garcia-

Gaines and Frankenstein, 2015).

[TABLE 2]

[FIGURE 4]

### 2.6.3. Calibration Parameters for Point Scale Evaluation

As with other hydrological models, the VIC model needs to be calibrated for optimized performance over a specific region. Table 3 shows VIC's key calibration parameters; more information on calibration parameters and methods can be found in past VIC studies (e.g. Elsner et al., 2010; Liang et al., 1994; Maurer et al., 2002). We used calibrated parameters determined

by Maurer et al. (2002) for each flux tower station (the last two columns of Table 3). We also tested the sensitivity of soil

moisture content, crop growth, and irrigation demand and losses to different calibration parameters using the ranges available in Column 3 of Table 3 and differences were negligible.

[TABLE 3]

### 2.6.4. Parametrization of Growth Stages in CropSyst

Thermal accumulation time in CropSyst is used to represent crop phenological development and the rate of biological activity (McMaster and Wilhelm, 1997). Specifically, the sum of growing degree days (GDD) is used to specify the time needed to reach specific phenological periods. We parameterized VIC-CropSyst for each site using published dates of crop growth stages (Table 4); meteorological information was used to convert calendar days to GDDs. Peak LAI was acquired from the MODIS LAI product (Cohen et al., 2006). Missing phenological information was estimated from the MODIS-derived peak LAIs as follows: i) flowering is 2-7 days after peak LAI, ii) filling starts 7-14 days after flowering, and iii) maturity happen 30-45 days into the filling period. Table 4 shows estimated/observed dates of the growing stages.

[TABLE 4]

### 2.6.5. Pacific Northwest Climate, Soil and Crop Information

We used the gridded historical climate data developed by Abatzoglou and Brown (2012), including precipitation, minimum and maximum temperature and wind speed (Table 5). Soil input file was developed using the STATSGO dataset (Schwarz and Alexander, 1995); to develop the soil file we used the same parameters as Elsner et al (2010) except we added the clay percentage because, as mentioned earlier, VIC-CropSyst uses Saxton et al (1986)'s Pedotransfer functions and can internally calculate the soil parameters (e.g. hydraulic conductivity, field capacity, bulk density). The calibration parameters (Table 3) used for simulation of ET over the Pacific Northwest were taken from Yorgey et al. (2011). Crop distribution information over the region was developed using the Washington State's Department of Agriculture for Washington State and the United State Department of Agriculture (USDA)'s cropping information for outside of the Washington State (Boryan et al., 2011). More information on crop types and crop input parameters (e.g. phonological periods, radiation use efficiency, transpiration use efficiency, maximum leaf area index, etc.) can be found in Barik et al. (2017), Hall et al. (2017) and Rajagopalan, et al. (in review).

TABLE [5]

## 3. Evaluation and Application

### 3.1. Point Scale Evaluation

#### 3.1.1. Applied Irrigation Water

Figure 5 compares recorded and simulated irrigation water (mean error=13%). Discrepancies may be due to reduction of crop yield in the field due to stresses that are not captured in the model such as impacts of weed or pests. Also, yields measured in small plots are subject to sampling uncertainty; In addition, simulated irrigation events are likely to include an extra event at the end of the season when irrigation managers stop irrigating earlier due to crop senescence.

[FIGURE 5]

#### 3.1.2. Evapotranspiration (ET)

Figure 6 depicts the comparisons between monthly simulated and observed ET over irrigated and non-irrigated sites. While the model tends to overestimate ET, particularly during the month with larger ET, simulations are more accurate at the NE irrigated site. Root mean squares errors (RMSEs) for the NE and IL stations were 8.0 and 1.0 (mm/day), respectively. In general, the deviation between observed and simulated ET is higher in the summer months. One explanation for this bias is that we do not consider the feedback of evaporative losses from irrigation droplets ($E_d$) and canopy-intercepted water ($E_c$) to the local microclimate system, while in reality these evaporative losses will lower ambient temperature and decrease vapor pressure deficit (VPD) (Kohl and Wright, 1974; Liu and Kang, 2006), thereby reducing irrigation demand. In the Biosphere-relevant Earth system model (BioEarth) project (Adam et al., 2014) this shortcoming is being addressed through coupling of VIC-CropSyst to atmospheric models. Inaccuracy of the meteorological data or uncertainties related to unrecorded management practices such as deficit irrigation can be other sources of error. This deviation can also be explained by a typical 20% systematic error in flux tower ET observations, which tend to underestimate the latent heat fluxes. This energy imbalance issue has been discussed in many studies by the microclimatological community (Frank et al., 2013; Leuning et al., 2012; Mahrt, 1998; Wilson et al., 2002).

[FIGURE 6]

### 3.1.3. Corn Yield

Figure 7 compares simulated and observed corn yield over the two sites. The mean error of simulated yield for NE (irrigated) and IL (non-irrigated) were 9% and 3%, respectively. Although Figure 7 does not show a systematic overestimation by the model, a combination of inaccurate meteorological data, missing processes (e.g. lack of VPD feedback as discussed in section 3.1.2) and unrecorded conditions such as insufficient irrigation water or heat stress can contributes to these discrepancies. The fact that the error is smaller over the non-irrigated site can be explained by the fact that irrigation management did not have to be simulated, thereby reducing the opportunity for introducing model error.

[FIGURE 7]

### 3.1.4. Soil Moisture

Figure 8 compares simulated and observed soil moisture over the two sites. Because the soil moisture sensors were placed at 10 and 25 cm depths a the NE site and at 2.5 and 10 cm depths at the IL site, we aggregated the first three VIC soil moisture layers (for a total thickness of 30 cm) for comparison against observations at the NE site. We compared just the first VIC soil moisture layer (10 cm depth) against observed at the IL site. The mean errors were 18% and 16% for the NE and IL sites, respectively. As with crop yield, soil moisture simulations are better for the non-irrigated site, particularly in terms of variability. The discrepancies may relate to the use of pedotransfer functions that convert soil textural characteristics to soil hydraulic properties (e.g. field capacity, permanent wilting point and hydraulic conductivity) for use in VIC-CropSyst (Pachepsky and Rawls, 1999; Tietje and Hennings, 1996). Also, scale discrepancies between the sensors' point-scale observation and the grid-scale simulation (Crow et al., 2012; Robinson et al., 2008) and inaccuracy of meteorological and soil data can be other sources of error. Additionally, imperfections in model processes such as soil water movement, evapotranspiration and irrigation loss calculation can contribute to the error.

[FIGURE 8]

### 3.1.5. Leaf Area Index (LAI)

Figure 9 shows that VIC-CropSyst is able to capture the magnitude and seasonality of observed LAI, with a slight underestimation of peak LAI. The information we used for calibration of phenological periods (Figure 9) are not specifically collected for the two study sites, but instead were based on state-scale studies and reports; this is a potential source of error in simulation of LAI. Because of limited information at flux tower sites, we did not consider all of the crop-related

parameters (e.g. radiation use efficiency, maximum crop coefficient and maximum crop coverage) during calibration, which can also lead to some discrepancies (e.g. Jalota et al., 2010; Klein et al., 2012).

[FIGURE 9]

### 3.2. Regional Evaluation of Evapotranspiration (ET)

We used VIC-CropSyst to simulate ET over the CRB portions of three states: Washington, Idaho and Oregon (Figure 10). Simulated ET was aggregated from the original model resolution of $1/16^{th}$ to 0.5 degree for comparison against the upscaled ET product derived from the FLUXNET eddy tower network (Baldocchi et al., 2001). Liu et al (2013) described the details

of the creation of the empirically-derived or "observed" ET map. They also compared the observed ET with an offline (from CropSyst) version of VIC-simulated ET and reported a systematic underestimation of simulated ET over warm irrigated areas. Our ET results show that VIC-CropSyst's simulated ET in general produces a lower error as compared to VIC-offline, especially over irrigated areas; error over irrigated landscapes was reduced from about 28% to 17%, a 40% reduction. However, it is important to note that another source of the discrepancy is due to inaccuracy of the observed ET product

because it was developed using a limited number of flux tower stations as well as empirical formulas that also have inherent errors (see Liu et al. 2013 for details).

[FIGURE 10]

**3.3 Regional Evaluation of crop yields and irrigation demands.**

Rajagopalan et al. (in review) performed an evaluation of county level aggregated irrigated crop yields against NASS crop yield statistics, and comparison of average modelled irrigation demands from the Columbia Basin Project area in the CRB to irrigation diversions. The mean annual yields between observed and simulated values are in agreement with relative errors less than +/-5%. On average, the model simulated annual irrigation demands where about 20% less than diversions. Part of

this difference can be explained by the fact that diversions account for conveyance and seepage losses in the distribution system.

## 4. Examples of VIC-CropSyst Application

### 4.1. Simulation of Agricultural Adaptation in Response to Climate Change

Farmers adapt their agricultural management to minimize unfavourable impacts of stressors such as climate change (Kurukulasuriya and Rosenthal., 2003). Possible agricultural adaptation strategies have been discussed (e.g. Anwar et al.,

2013; Howden et al., 2007; Kurukulasuriya and Rosenthal., 2003; Smit and Skinner., 2002; Smith et al., 2000). However, lack of appropriate simulation tools to assess the effectiveness of an adaptation decision while capturing complex regional impacts is a significant shortcoming. VIC-CropSyst simulates common adaptation strategies used by farmers, and captures the consequences of these adaptation strategies on local and regional hydrology and land-atmosphere interactions. Table 6 shows a list of adaptation decisions that can be handled by VIC-CropSyst. These decisions range from short-term tactical (T)

to long-term strategic (L) decisions.

[TABLE 6]

### 4.2. Foundation for Integration within Other Modeling Platforms

VIC-CropSyst can be used with other modeling frameworks such as atmospheric, socio-economics, and water storage and routing models. These integrations may simulate the human-land-climate nexus and provide scientists, stakeholders and policy makers with a broader understanding of the interactions and feedbacks in this nexus. VIC-CropSyst has been already used and implemented in various projects. Examples of these implementations are as follows. VIC-CropSyst can also be

used to investigate the hydrologic and atmospheric impacts of these adaptation decisions.

#### 4.2.1. Water Resources Management and Socio-Economic Studies

VIC-CropSyst  is used in conjunction with reservoir operation models in the CRB, and accounting for the process of water rights curtailment under shortages in Washington State and farmer response to curtailment, to identify the indirect impacts

on climate change on agricultural production though changes in water availability (Rajagopalan et al., in preparation). The current version of VIC-CropSyst (v2, as described herein) was also used in the most recent Columbia River Basin water supply and demand projection for the 2030s (Barik et al., 2017; Hall et al., 2017). These water supply and demand studies were submitted to the Washington State Legislature in the years of 2011 and 2016 and provide detailed information for each watershed in eastern Washington to the entire CRB as a whole. This information is being used by the Legislature for long-

term water supply planning.

VIC-CropSyst has been used to investigate different scenarios for renegotiation of the Columbia River Treaty (Rushi et al. 2017). Existing modeling efforts to date have focused primarily on the impact that treaty renegotiation would have on flood

risk, hydropower generation, and environmental flows (Cosens, 2010; Hamlet and Lettenmaier, 1999a); assessment of the impact of CRT changes on irrigated agriculture along the Columbia Mainstem is a knowledge gap. Rushi et al. (2017), therefore applied VIC-CropSyst linked to ColSim to simulate the complex impacts of climate change and the Columbia River Treaty on hydrology and agriculture in the river basin and concluded that climate change i) shifts water supply towards earlier in the season, ii) reduces flood risk in the upper CRB while increases frequency and magnitude of floods in the middle and lower parts of the basin, iii) shifts water demand towards earlier in the season in some locations with mixed effects on water rights curtailment risk, and iv) reduces hydropower generation. The authors found that the considered CRT scenarios can improve power generation and agricultural water demand while preventing floods in an altered climate.

VIC-CropSyst is an effective tool for studying the large-scale aggregated impacts of local management decisions and phenomena. For example, VIC-CropSyst was applied by Malek et al. (in review) who found that climate change-induced increases in evaporative (consumptive) losses from irrigation systems and decreases in non-evaporative irrigation losses (i.e., runoff and deep percolation) would lead to a decrease in reusable return flow, which would negatively affect basin-wide water availability and productivity.

VIC-CropSyst has also been used over the Yakima River basin (YRB) to evaluate the impacts of climate change on decisions related to investment in irrigation technology (Malek et al., 2016; in prep.). Economic damages of future more frequent droughts (Vano et al., 2010) are considered the main incentive to invest in more efficient irrigation technology (Berger and Troost, 2014). To analyze future changes in regional irrigation patterns, Malek et al. (in prep.) used VIC-CropSyst in conjunction with an economic model and the RiverWare model (Zagona et al., 2001). Figure 11 shows a result of this integration to simulate historical (1981-2006) drought frequency and severity, and the percentage of the YRB's perennial crop growers who are simulated to switch to more efficient irrigation systems to minimize the negative consequences of droughts during the two decades of 1990-2000 and 2050-2060. Also, any changes in agricultural activities (e.g., switching to a new irrigation system) directly impacts the hydrology of agricultural fields, thus changing return flow timing and magnitude and the availability of water for downstream users; these downstream consequences can also be simulated by this modeling platform. This is an example of how the human-land-climate nexus can be captured through a modeling framework that simulates large-scale hydrologic processes and regional water availability in a highly cultivated basin, while capturing the dynamics of farm-level irrigation decisions.

[FIGURE 11]

### 4.2.2. Land-Atmosphere Interactions

Irrigation and other agricultural decisions modify local to regional climate through changes in land surface conditions such as temperature, water vapor content and albedo (Fernández et al., 2001; Liu and Kang, 2006). This phenomena can be used

to compensate the negative impacts of heat stress (Lobell et al., 2008), which will be especially important in the future if there are more severe and frequent extreme events related to climate change (Long and Ort, 2010). These management decisions will also impact the regional water cycle, potentially leading to disruption in water availability (Adamson and Loch, 2014) and modifying fluxes of water to the atmosphere (Pielke Sr. et al., 2007). As a part of the BioEarth platform

(Adam et al., 2014), VIC-CropSyst is being coupled to an atmospheric model, the Weather Research and Forecast model (WRF; Michalakes et al., 2005; Skamarock et al., 2008) that can be used to quantify the impacts of irrigation and other agricultural management on atmospheric processes, as well as to assess how irrigation management can be used to mitigate heat stress.

## 5.  Conclusions

Meeting future food demand will require an extensive understanding of the interactions between agricultural and other systems, such as water resources planning and management, socioeconomic, and atmospheric processes. The main purpose of this study was to develop the VIC-CropSyst platform that provides tightly-integrated and mechanistic representation of both cropping systems and water/energy cycles at regional to global scales. Tight integration between VIC and CropSyst

necessitated modification of both models, including how the models handle soil movement and vertical distribution, transpiration, LAI, and irrigation. Evaluation of VIC-CropSyst over two flux tower sites shows that the coupled model captures key agronomic and hydrologic states and fluxes at the field scale. Furthermore, implementation of VIC-CropSyst over the U.S. Pacific Northwest region reduced ET simulation error by 40% over irrigated landscapes.

The VIC-CropSyst platform enables the land surface modeling community to investigate a variety of agricultural management decisions, including crop choice, planted acreage, planting and harvesting date, and multiple irrigation management options. In particular, the new mechanistic irrigation model, which is tightly coupled with both the energy and water cycles, can be used to address questions related to the interaction of climate, hydrology, river basin water management, and irrigation management strategies.

VIC-CropSyst can be integrated with different modeling platforms to capture the dynamics of the human-land-climate nexus. This can potentially improve the understanding of environmental processes in highly-cultivated basins and can be used to investigate best management practices to promote future sustainability of agricultural production while preserving water resources and minimizing the negative intended and unintended consequences of human actions. Some examples of these

implementations are as follows:

*Coupling with water resource management and socioeconomic models:* This involves simulating regional water availability and agricultural productivity, adaptive responses of farmers to climate change, and unintended consequences of these adaptation decisions.

*Coupling with weather and climate models:* VIC-CropSyst will also provide capabilities to investigate the dynamics of agricultural management decisions on local to regional weather and climate patterns through modifications of energy and water fluxes (Barnston and Schickedanz, 1984; Douglas et al., 2009; Kohl and Wright, 1974). This promotes the understanding of, for example, how irrigation management and technology can control negative impacts of heat and water stresses on crop yield.

VIC-CropSyst is being used in earth system models (EaSMs) such as BioEarth (Adam et al., 2014) and can be implemented in other EaSMs such as the Platform for Regional Integrated Modeling and Analysis (PRIMA; Kraucunas et al., 2014). Implementation of VIC-CropSyst in EaSMs facilitates a powerful representation of large-scale interactions between different biophysical and socioeconomic components over areas with significant agricultural activities. This is a transformational step in the understanding of the food-energy-water nexus which can lead to efficient and more sustainable management decisions that co-balance and benefit all three sectors.

**Code and/or data availability**

The VIC-CropSyst is a freeware open source community model; source codes, user manual and test cases will be distributed through contact to Keyvan Malek (keyvan.malek@wsu.edu), Jennifer Adam (jcadam@wsu.edu) and Mingliang Liu (mingliang.liu@wsu.edu).

**Acknowledgments**

We would like to express our gratitude to two anonymous reviewers for their comments that improved this manuscript. This research was funded by Washington State Department of Ecology's Office of Columbia River Basin and the Department of Agriculture, National Institute of Food and Agriculture Grant Number 2011-67003-30346 (Biosphere Relevant Earth System Model; BioEarth). This research is also financially supported by Washington State University's Graduate School.

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

**Tables**

**Table 1- Impacts of climate change on crop yield, as discussed by Kurukulasuriya and Rosenthal, 2003; Leakey et al., 2009; Reilly, 2002; Rosenzweig et al., 2001; and Rowan et al., 2011.**

| Impact factors | Mechanism of impact | Direction of impact on yield | References |
|---|---|---|---|
| $CO_2$ concentration | More efficient photosynthesis | + | Kurukulasuriya and Rosenthal, 2003; Leakey et |

| | | | al., 2009 |
|---|---|---|---|
| | Crop water use efficiency | + | Leakey et al., 2009 |
| | Nutrient use efficiency | + | Ainsworth and Rogers, 2007 |
| Temperature | Crop growing period length | - | Kurukulasuriya and Rosenthal, 2003; Dukes and Mooney, 1999 |
| | Planting date | +/- | Parry et al., 2005 |
| | Timing and rate of crop growth and phenology | +/- | Tao et al., 2003 |
| | Pest and weed growth and development | - | Kurukulasuriya and Rosenthal, 2003 |
| | Fruit quality | - | |
| Humidity | Changes in stomata functioning | + | Leakey et al., 2009; Nijs et al., 1997 |
| Precipitation | Changes in soil moisture and irrigation water resources | -/+ | Rowan et al., 2011 |
| Frequency of climate extreme events (droughts and heat waves) | Crop productivity | - | Rosenzweig et al., 2001 |
| Temperature + | Water availability for | - | Adam et al., 2009; |

| Precipitation | irrigated agriculture over snow dominant basins | Barnett et al., 2005; Elsner et al., 2010; Mote et al., 2005 |
|---|---|---|

**Table 2 – Two flux tower stations used for evaluation of the VIC-CropSyst. Nebraska site is irrigated using a center pivot system and the Illinois flux tower station is rain-fed.**

| Stations | State | Irrigated | Cropping pattern | Period | Soil type | Average Precipitation (mm) | Average Temperature (°C) |
|---|---|---|---|---|---|---|---|
| Mead Irrigated | Nebraska (NE) | Yes | Corn | 2001-2008 | Silty clay loam | 789 | 10.1 |
| Fermi National Laboratory | Illinois (IL) | No | Corn/ Soybean | 2002-2007 | Silty clay loam | 929 | 9.2 |

**Table 3–Calibration parameters used for VIC-CropSyst over the two study sites (columns 5 to 6) and over the Columbia River Basin (CRB). Column 3 represents the ranges of these parameters used for the sensitivity studies.**

| parameter | Description | Range | CRB | NE | IL |
|---|---|---|---|---|---|
| bs | Adjusts partitioning of precipitation to | 0.001-0.4 | 0.1-0.3 | 0.2 | 0.31 |

| | | | | |
|---|---|---|---|---|
| | runoff and infiltration | | | |
| Ds | Base flow parameter- fraction of base flow parameter | 0.001-0.99 | 0.001-0.88 | 0.005 | 0.72 |
| Ws | A fraction of maximum base flow indicating where the base flow curve starts | 0.4-0.9 | 0.51-0.91 | 0.8 | 0.53 |
| Ds-Max | Maximum daily base flow generation | 0.1-30 | 0.2-10 | 10 | 28.61 |

**Table 4- Estimated calendar days correspond to each of the growing stages in two study sites. Some of the information is from references listed for each site.**

| | crop type | planting | emergence | peak LAI | flowering | filling | maturity | reference |
|---|---|---|---|---|---|---|---|---|
| NE | corn | 127 | 140 | 195 | 205 | 225 | 255 | (Sakamoto et al., 2010) |
| IL | corn | 125 | 137 | 200 | 208 | 212 | 250 | (Nafziger, 2013) |

**Table 5- Soil, climate, vegetation and crop information used for regional evaluation of VIC-CropSyst over the U.S. Pacific Northwest. The resolution of the input data was 1/16th °.**

| Input | Source | Information used by VIC-CropSyst |
|---|---|---|
| Weather | Abatzoglou and Brown (2012) | precipitation, minimum and maximum temperature and wind speed |
| Soil | STATSGO (Schwarz and Alexander, 1995) | latitude, longitude, sand and clay content, hydraulic conductivity, field capacity, bulk density, etc. |
| Crop/Vegetation | USDA/WSDA vegetation distribution maps (Boryan et al., 2011; Yorgey et al., 2011) | crop type, acreage, irrigation systems, etc. |

**Table 6- Summary of adaptation strategies that can be handled by VIC-CropSyst: the modeling platform is able to simulate the impacts of local decisions on agricultural productivity and at the same time capture the impacts of these decisions on regional land-atmospheric interactions and hydrological water availability in the basin.**

| | Adaptation strategy | Timing* | Duration** |
|---|---|---|---|
| 1 | **Crop-related adaptation strategies** | | |
| | i- Crop choice and rotation | R* | L |
| | ii- Cropping acreage and location of cropping activities | R and A | L |
| | iii- Timing of planting and harvesting date | C and A | T |
| | iv- Using new variety of the same crop | R | L |
| 2 | **Long term strategic water management adaptations** | | |
| | i- Irrigation system or nozzle | R and A | L |
| 3 | **Seasonal adaptations to respond to altered water deficit and temporal availability of water** | | |
| | i- Deficit irrigation magnitude | C | T |
| | ii- Deficit irrigation timing in a season | C | T |
| 4 | **Short term tactical adaptation to minimize the impacts of heat stress** | | |
| | i- Supplementary/over irrigation | | T |
| | ii- Irrigation frequency | C | T |
| | iii- Irrigation intensity | C | T |

*According to Smit and Skinner (2002), the timing of adaptation decision can be A-Anticipatory (proactive), C-Concurrent (during) or R-Responsive (reactive)

**Duration of adaptive actions can be short term-tactical (T) and long-term strategic (L)(Smit and Skinner, 2002)

**Figures**

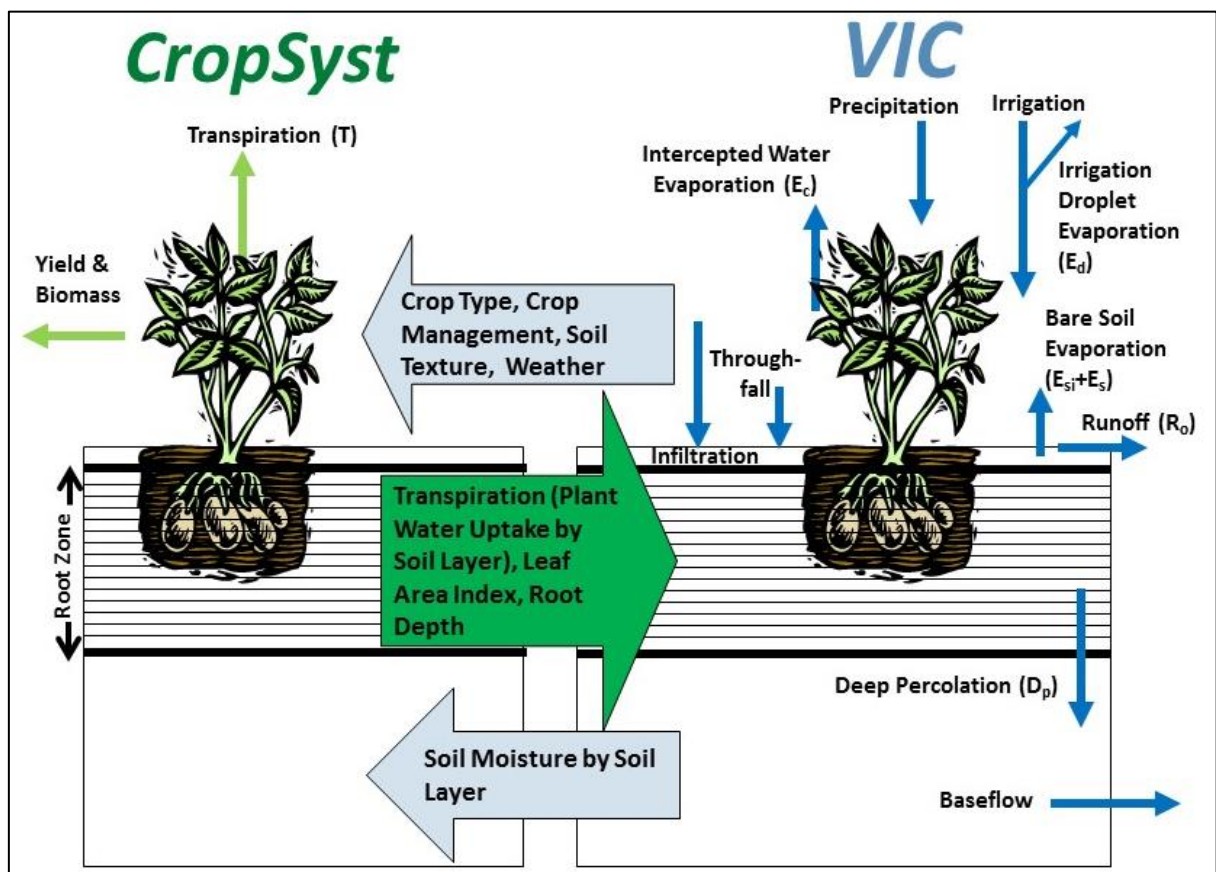

**Figure 1- This schematic shows how VIC and CropSyst are coupled. VIC provides the availability of water and energy to CropSyst. CropSyst uses this information to grow the crop, produce biomass and yield, and simulate transpiration. CropSyst passes back the information that is needed by VIC (e.g., the distribution of transpiration uptake in different soil layers, LAI, and root depth) to simulate the hydrologic and energy cycle and the scheduling of irrigation.**

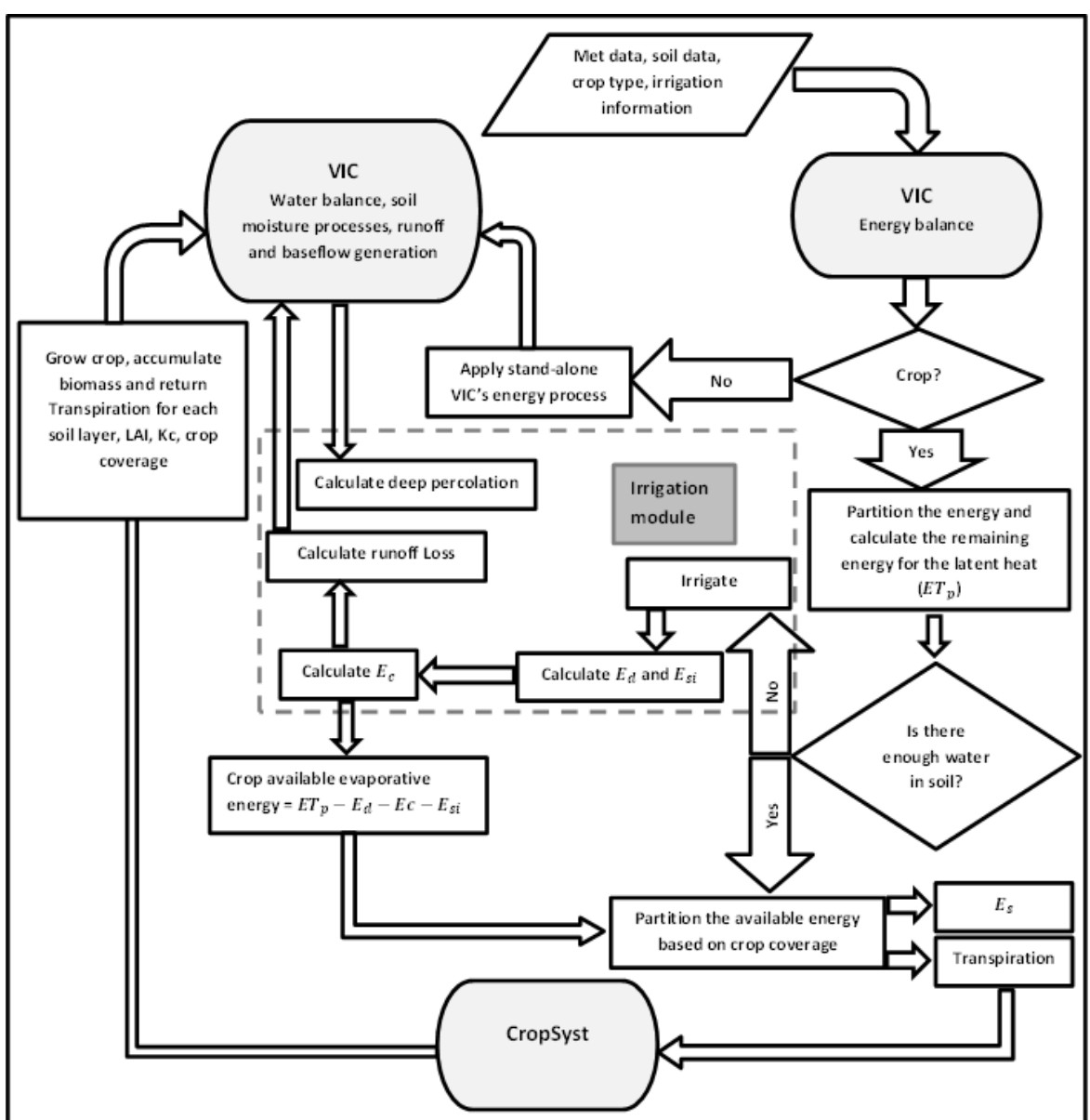

**Figure 2 – Algorithm used in VIC-CropSyst to partition available energy into different evaporative components. The energy and water balances are handled by the VIC model. CropSyst receives the amount of energy available for transpiration and the availability of water in the soil to determine crop water uptake. VIC needs actual transpiration in different layers of the soil to close the water cycle. Communication between the two models happens for every time-step.**

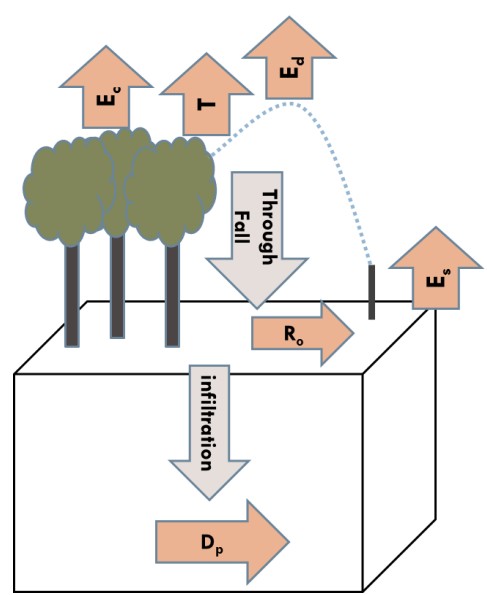

**Figure 3** –Pathways of irrigation water loss simulated in the irrigation module.$E_d$: evaporation from irrigation droplets, $E_c$: evaporation from irrigation water intercepted by canopy, $E_s$: evaporative loss from soil surface, $D_p$: Deep percolation loss and $R_o$: Irrigation runoff loss. The efficiency of irrigation water is calculated by considering total applied water and all loss terms.

$$Ef = 100 \times \left( 1 - \frac{E_d + E_s + E_c + R_o + D_p}{total\ irrigation\ water} \right)$$

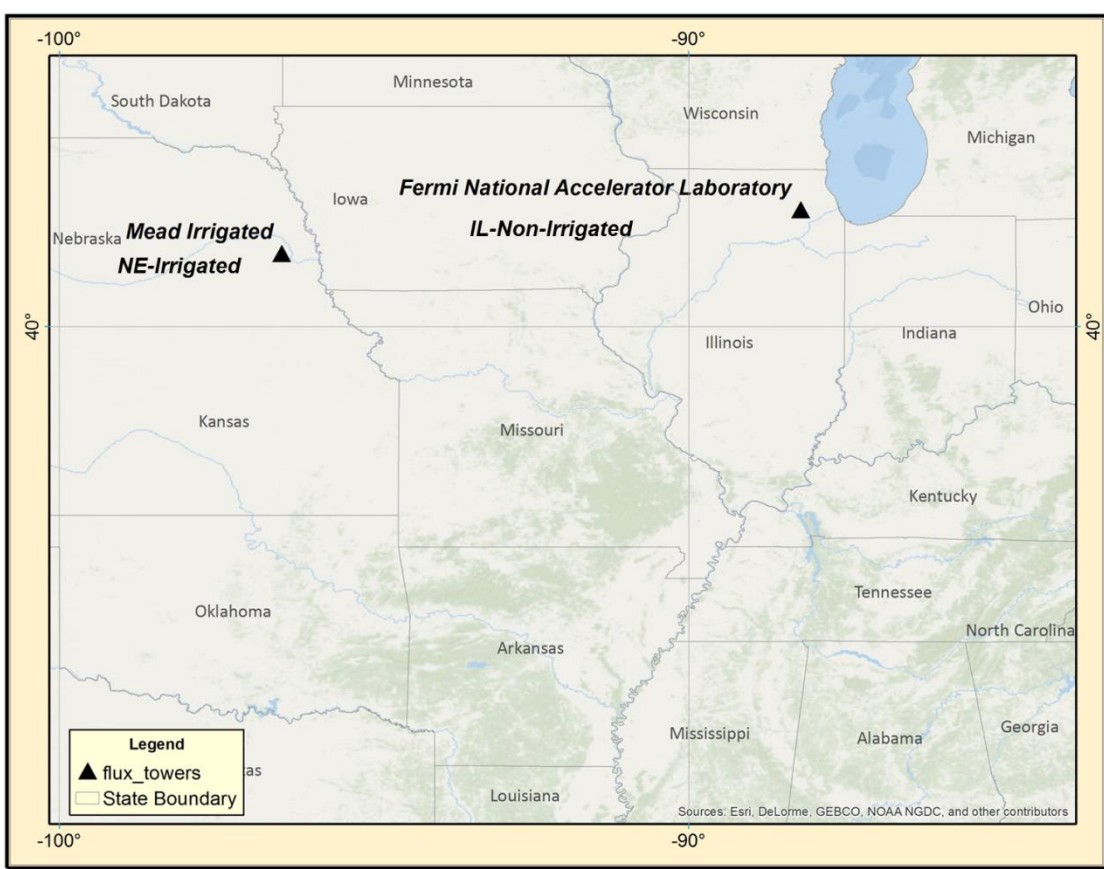

**Figure 4 - Location of the two flux tower sites in the U.S. two flux tower sites are all in agricultural fields. Mead irrigated site (NE) is located in the Nebraska; Fermi National lab site (IL) is located in the Illinois; NE is irrigated and the IL is a non-irrigated agricultural site.**

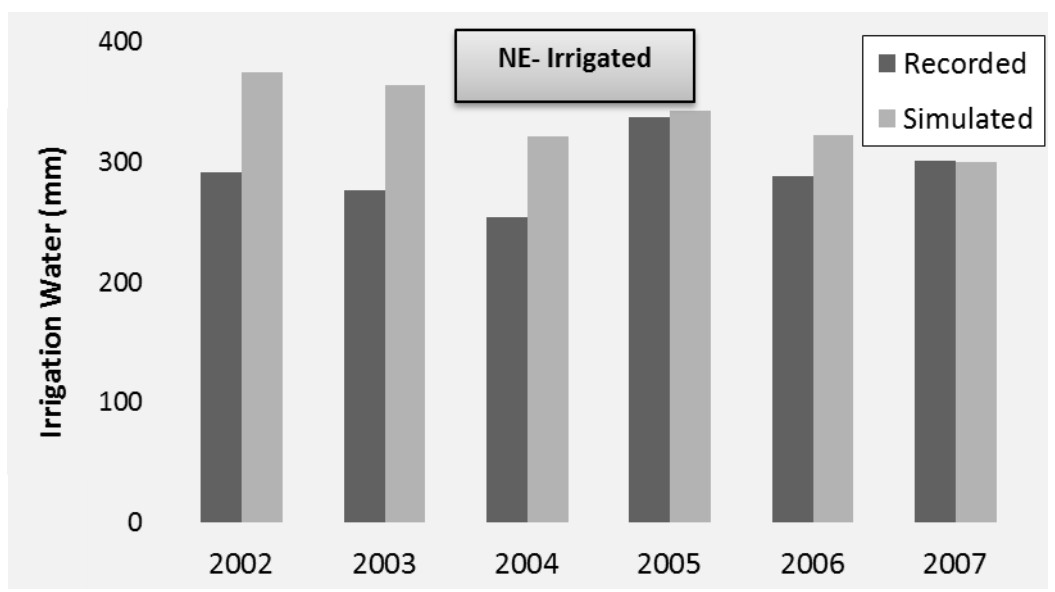

**Figure 5 - Simulated versus recorded irrigated water in an irrigated corn field at the NE flux tower site.**

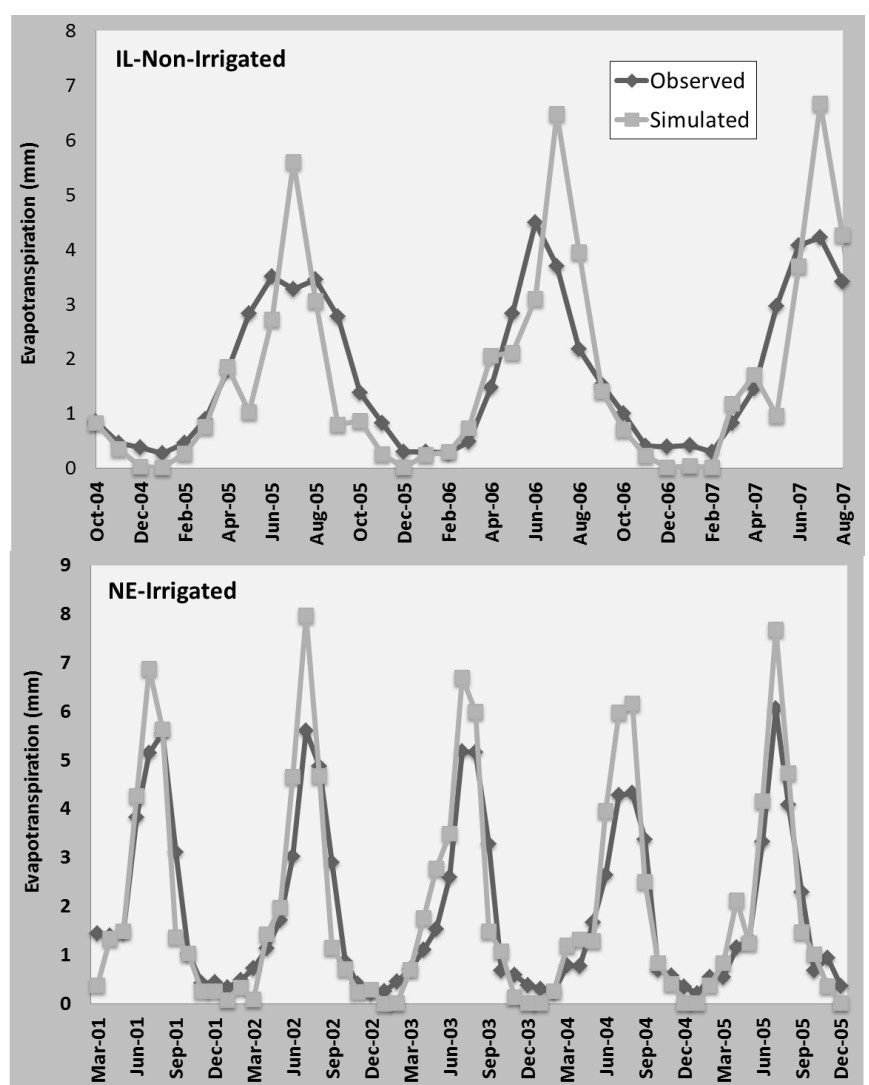

**Figure 6- Comparison of simulated and observed corn evapotranspiration (ET) at two flux tower sites located in NE and IL. The NE site is irrigated while IL is a non-irrigated field.**

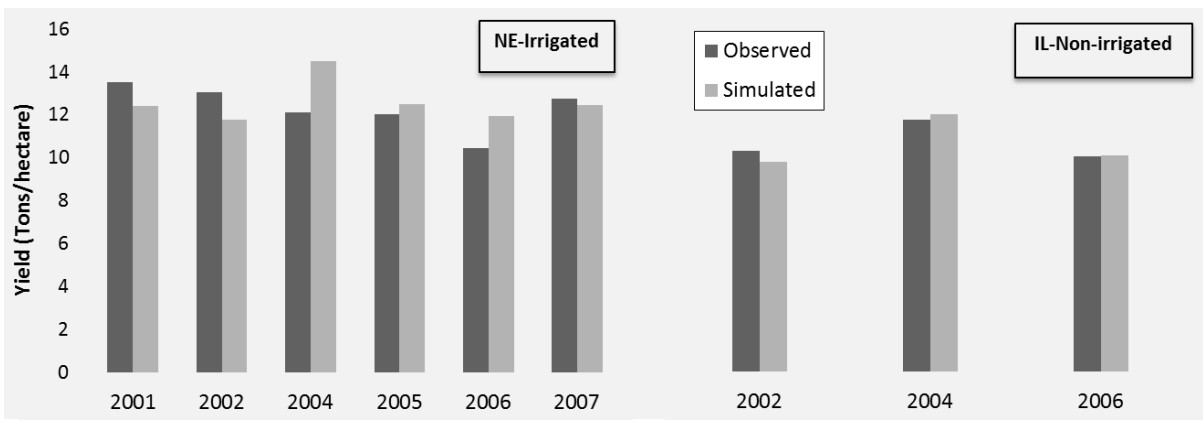

**Figure 7 – Comparison of simulated and observed corn yield at two flux tower sites for the years during which yield observations were taken.**

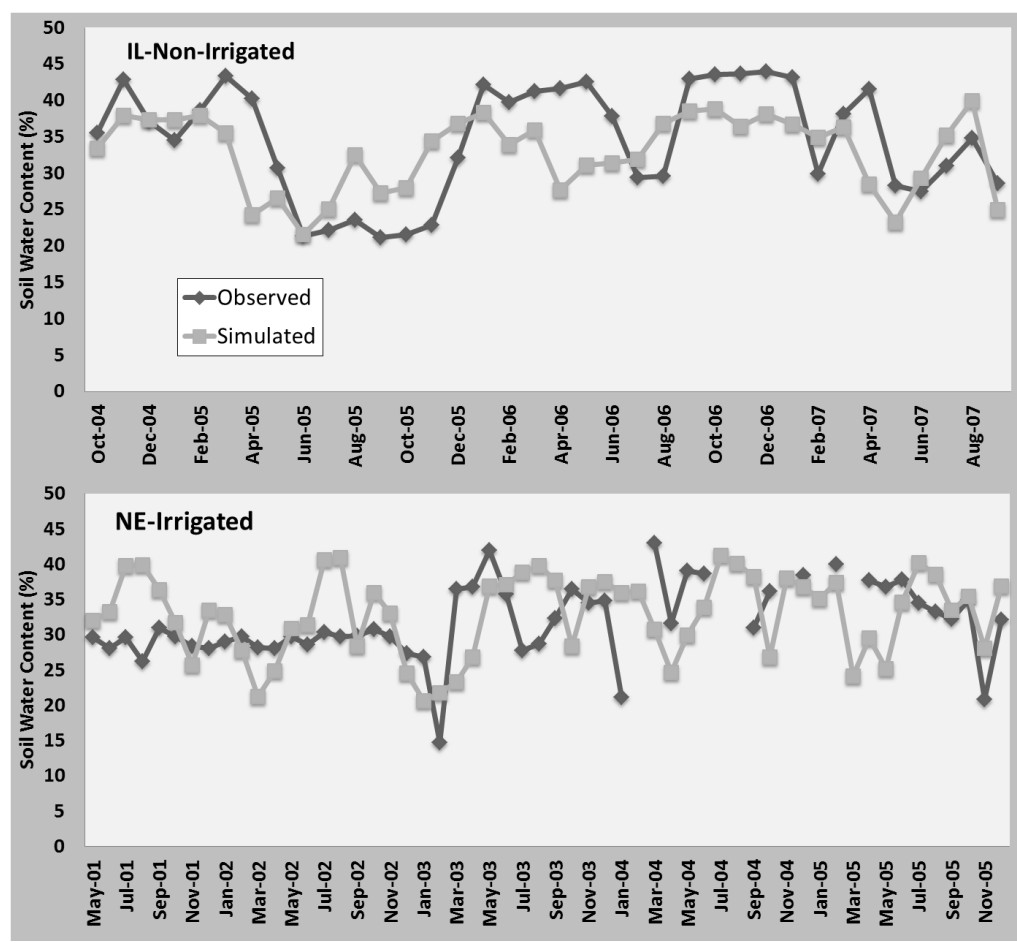

**Figure 8- Comparison of simulated and observed soil moisture at the flux tower sites located in IL (top) and NE (bottom).**

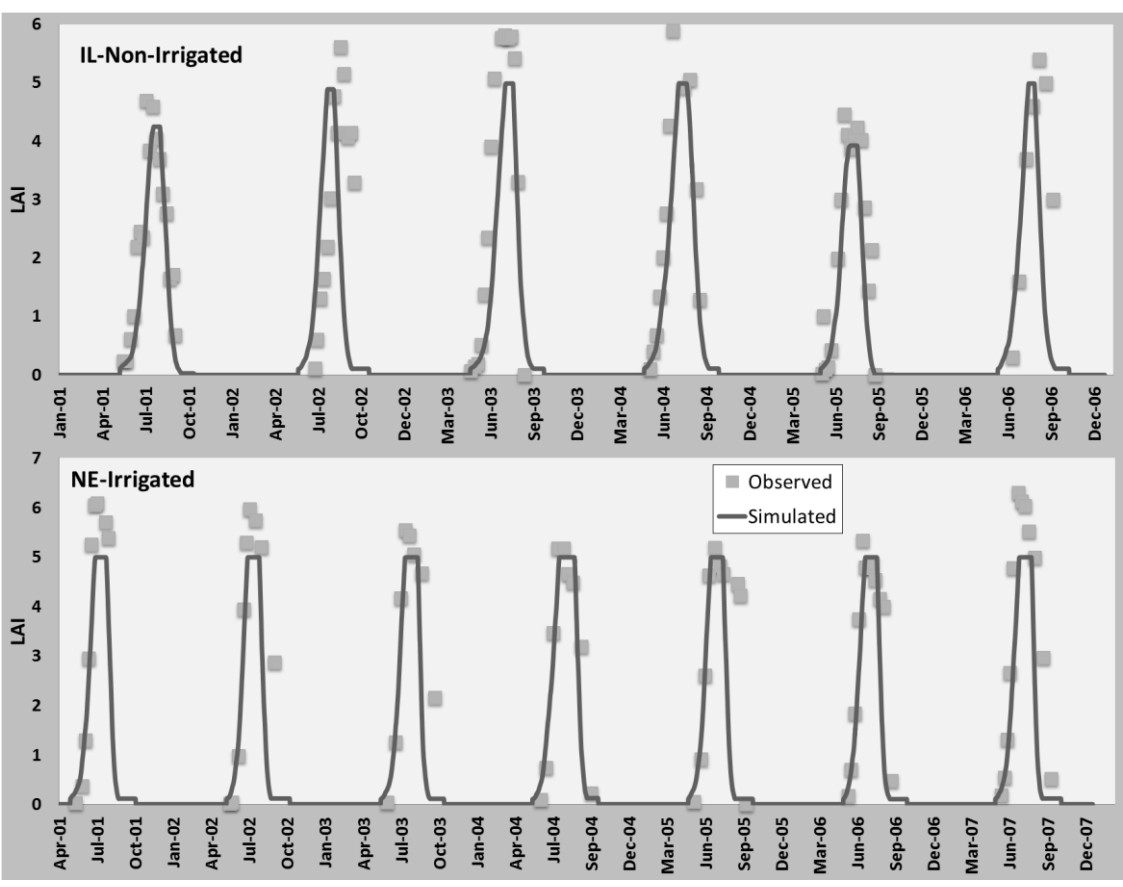

**Figure 9- Comparison of simulated and observed corn LAI over two flux tower sites located in IL (top) and NE (bottom).**

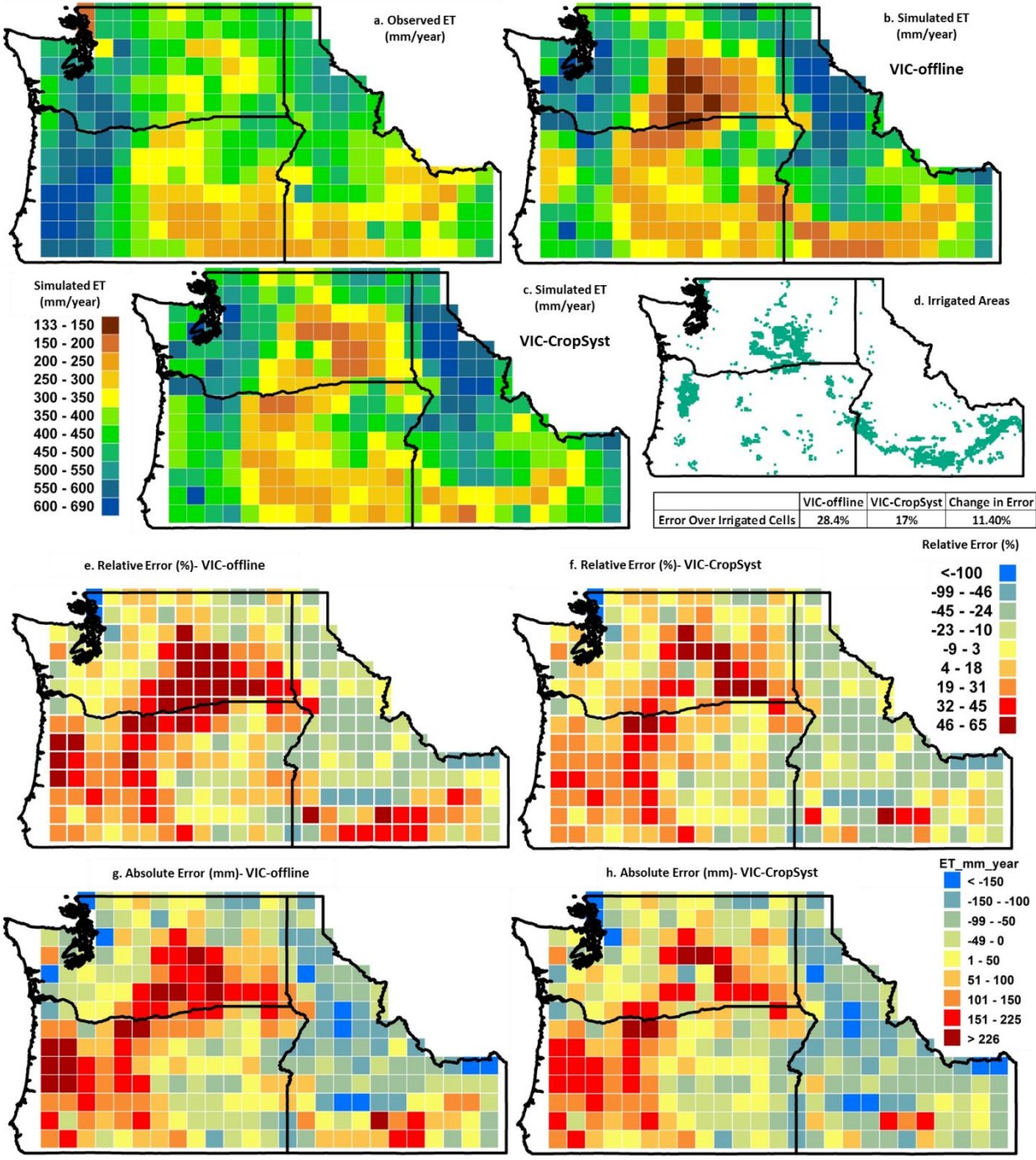

**Figure 10 – Comparison of simulated and empirically-derived ET over the U.S. Pacific Northwest. The simulation and observation period is 1982-2008.**

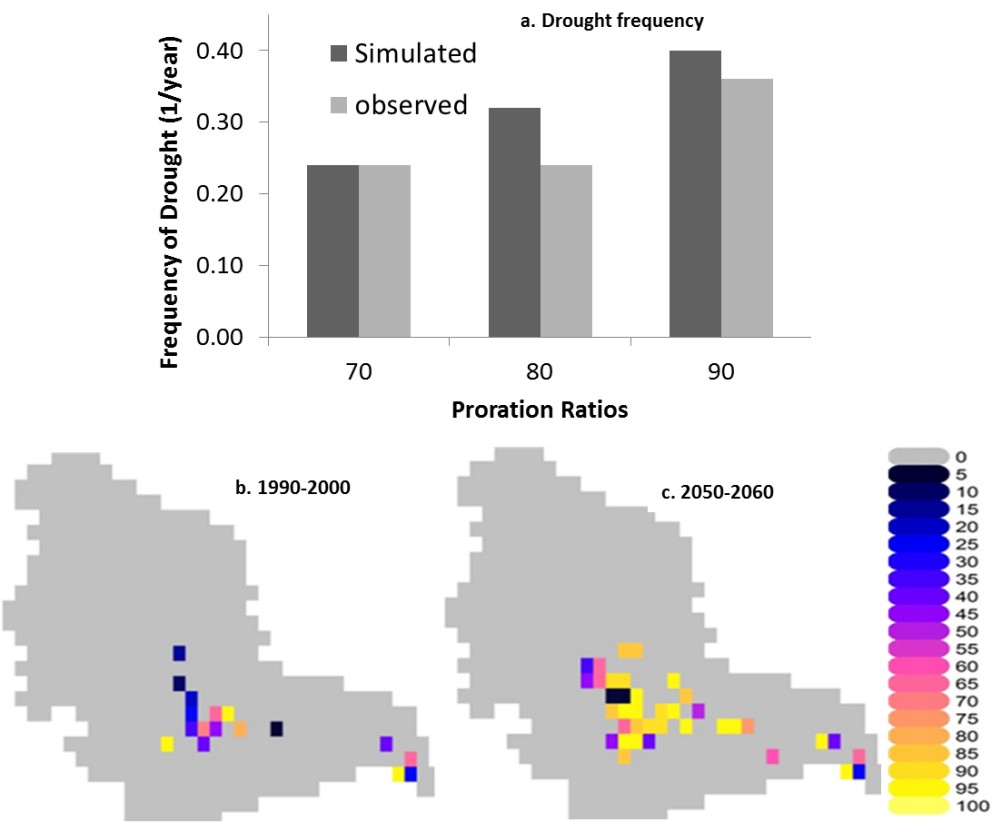

**Figure 11- Regional application of VIC-CropSyst in conjunction with a river system model (YAK-RW; (Hubble, 2012; Zagona et al., 2001) and an economic model to simulate historical (1981-2006) drought frequency (panel a), when the percentage of the water right allocated for the irrigation season (i.e., proration rate) is lower than 70%, 80% and 90%. Panels b and c (Malek, K et al., 2016) show the percentage of farmers (perennial crop growers) who invest in new efficient irrigation technologies in response to simulated droughts during the two decades of 1990-2000 (panel b) and 2050-2060 (panel c).**