# Peer review of "VIC-CropSyst-v2: A regional-scale modeling platform to simulate the nexus of climate, hydrology, cropping systems, and human decisions"

_Geoscientific Model Development, 2016_

## Author Comment (AC1) · 5 Jan 2017

Dear Dr. Kerkweg,

Authors would like to thank you for your comment, we will make sure that the versioning information is included in our revised submission.

Regards, Keyvan Malek
* * *

---

## Referee Comment (RC1) · Anonymous Referee #1 · 23 Feb 2017

**Review of the paper**

"VIC-CropSyst: A regional-scale modeling platform to simulate the nexus of climate, hydrology, cropping systems, and human decisions" by Malek et al.

In this paper the authors describe a simulation platform that captures the nexus of land, atmosphere, and human processes in one model. To this end, they have coupled the macroscale Variable Infiltration Capacity (VIC) hydrologic model and the CropSyst agricultural model.

The paper is well written, good to understand and the results are well described. The topic of the study is of interest for scientists and natural resources decision makers.

However, there are some shortcomings, and the major in my view is that they ignore the huge amount of literature and work which already has been done in this direction. The authors present the topic of the study, to fully couple hydrological and agricultural models in one system considering feedbacks, as if this is an entirely new field. Cited are only global scale studies with comparable approaches but not so far developed. But the case studies given in this paper are at the regional and even local scale. And at the regional scale, first attempts to couple hydrological and crop models started already in the late 70ties. A prominent example is SWAT (Soil and Water Assessment tool), nowadays also applied at the continental and global scale, and many other exist. At the global scale, the models ORCHIDEE and LPJmL have coupled water and crop modules etc.

Minor comments:

Page 3, last para: How do you define return flow?

Page 4, second para: "...the current state of LSMs is not capable of capturing agricultural processes in a detailed manner". However, the literature cited is mostly older than 2010, and the most recent 2014. This is not the current state.

Page 6, first para: Does VIC consider reservoirs and other water management measures?

Page 10, last para: "As with other hydrological models, the VIC model needs to be calibrated ...". This is only part of the story: State of the art is to calibrate AND validate in a split-sample approach. So, are the results shown in Figures 5-10 from the calibration or from the validation period? If, for example, the results in Figure 7 are from the calibration period, I would expect them to be good.

---

## Referee Comment (RC2) · Anonymous Referee #2 · 10 Mar 2017

In this paper the authors present a regional version of a coupled model system, the VIC hydrological model and the CropSyst crop model. The objective of the coupled system is a.o. to evaluate the potential impact of adaptation measures taken by farmers on basin scale hydrology.

General comments

- The authors could elaborate a little more on the potential applications of this coupled model system, as they make not clear what is the added value of the coupled system versus the individual models. - The authors claim that the coupled model system can be used to evaluate the impact of certain agriculture related adaptation measures over the region or river basin, but I was surprised to see that this impact is only modelled

in one way. The way I understand the model from this manuscript, is that irrigation water is assumed to be always available, but the source of this irrigation water is not discussed. Unless water is always extracted from deep confined groundwater layers, there should be an effect of water withdrawals for irrigation on streamflow and water availability downstream. Since VIC explicitly calculates streamflow, I think it is a missed opportunity not to include this interaction, especially since irrigation withdrawals have been implemented before eg. by Haddeland et al. To my understanding there is no consideration of water shortage for irrigation. - I miss the broader embedding of this research in the existing body of knowledge. This model is certainly not the first to combine a hydrology and crop model (eg. LPJmL), but the authors seem to mainly relate to their own research in the introduction. - For sake of reproducibility, the authors should include more background of the models and equations used.

Specific comments

- The abstract would benefit from a little more text on potential application of this model, and more specific on how it can be used to inform 'policy and best management practices to promote sustainable agriculture'. What can the model do, that cannot be done without a model? - Pg 2, r 6.. there are unanswered questions.... what are the unanswered questions and how are you going to address them? - Pg2. r 6. The consequences of what kind of decisions are not understood? Can you give an example of a situation where that happened and where the use of this model could have helped? - P2.r7. What are the knowledge gaps? - P3. R 7. What kind of human decisions? - P3 r 22. Can you give examples of the management decisions farmers can make? - P4 r1. What is meant by 'large scale results'? - P5. I am surprised to see that you are not referring to other models that are also capable of relating hydrology to crop production. - P5. Could you explain a little better why the (vertical) soil water balance of VIC is better than the one that was originally included in CropSyst? Since you are not using the lateral flow generated in VIC, the advantage of this coupling is not completely clear (to me). - P6. L14. It would be good to have a little more information on the crop model,

since the information given here is very limited. E.g. which crops are included, how are sowing and harvest dates determined, is there any management included, how is yield calculated. - P8. L6. As long as the paper describing the irrigation module in more detail is unpublished, it is difficult to judge the model, so a little more detail regarding algorithms is required here. - P8. It is very impressive that the model is able to simulate over 40 different irrigation systems, but it would be good to briefly descibe how differences between those systems are implemented in the model and which assumptions are made. - P8 l22. Which crops are included? - P8 l29. For readibility, it would be good to write out the meaning of Esi in this sentence and put Esi between brackets. - P9 l8. What is the equation, I think it would be good to add it here. - Idem for the equations in line 12 and line 14 (referring to an equation in an article in preparation is not ideal). - P10 l5. The simulated variables that are compared, could results been shown for all the mentioned variables? - P10 l18. The soil files were modified using available information, could this be explained? - P11, l 19. Are those climate data the same as the climate data mentioned on p 10 (DAYMET)? This is somewhat confusing. - P 11 line 25. I understood from table 2 and different figures that simulations were made for corn only, how was the crop distribution information used? - P12 section 3.3.1 Is water shortage for irrigation not considered at all? Could that be an issue in this region? - P12 section 3.1.2. Could the overestimation of etp also have to do with water shortage, the used crop parameterization? - P12 3.1.3 could you describe a little better how yields are calculated and also reflect on why the variability for irrigated yields is not captured. - P13 3.1.4 It would be interesting to see some reflection on the meaning of errors. - P14 I think it is good to emphasize here that the model can be used to evaluate the cumulative effects of large scale implementation of selected adaptation strategies over a basin or watershed.

---

## Author Response (AR1)

**Major comments:**

**"VIC-CropSyst: A regional-scale modeling platform to simulate the nexus of climate, hydrology, cropping systems, and human decisions" by Malek et al. In this paper the authors describe a simulation platform that captures the nexus of land, atmosphere, and human processes in one model. To this end, they have coupled the macroscale Variable Infiltration Capacity (VIC) hydrologic model and the CropSyst agricultural model. The paper is well written, good to understand and the results are well described. The topic of the study is of interest for scientists and natural resources decision makers.**

**However, there are some shortcomings, and the major in my view is that they ignore the huge amount of literature and work which already has been done in this direction. The authors present the topic of the study, to fully couple hydrological and agricultural models in one system considering feedbacks, as if this is an entirely new field. Cited are only global scale studies with comparable approaches but not so far developed. But the case studies given in this paper are at the regional and even local scale. And at the regional scale, first attempts to couple hydrological and crop models started already in the late 70ties. A prominent example is SWAT (Soil and Water Assessment tool), nowadays also applied at the continental and global scale, and many other exist. At the global scale, the models ORCHIDEE and LPJmL have coupled water and crop modules etc.**

First and foremost, the authors would like to thank Reviewer #1 for this constructive feedback of the article.

The authors thank the reviewer for this insightful comment. The authors agree that the incorporation of agricultural processes in hydrologic models such as SWAT dates back to the early stages of computer models, and this was not acknowledged in the manuscript. VIC and CropSyst are well-established large-scale hydrology and cropping systems models, respectively. The original intent was to present VIC-CropSyst as a contribution to large-scale land surface models due to its versatility and combined mechanistic simulation of crop and agricultural management processes as well as hydrologic processes. To address the reviewer's concern, in the revised manuscript, we incorporated a more comprehensive literature review that takes various types of hydrologic models into account.

Added/Modified

[revised manuscript text omitted]

**Answer to minor comments:**

**Page 3, last para: How do you define return flow?**

In this study, we are referring to the USBR (2010) definition of return flow as a non-evaporative, reusable loss of water through conveyance systems and the field-level application of irrigation water. In the revised manuscript, we added some explanation to clarify this return flow definition.

Added/Modified:

*In many agricultural basins, the availability of water for downstream users depends greatly on the return flow from upstream lands, which mainly .comes from non-evaporative, reusable loss of water through conveyance systems and field-level application of irrigation water.*

**Page 4, second para: ": : :the current state of LSMs is not capable of capturing agricultural processes in a detailed manner". However, the literature cited is mostly older than 2010, and the most recent 2014. This is not the current state.**

Authors' response: In our new submission, we have done an up-to-date literature review and added more information about recently published works on this topic.

**Page 6, first para: Does VIC consider reservoirs and other water management measures?**

Authors' response: While VIC-CropSyst does not simulate reservoir directly nor the potential management decisions in operating these waterbody compartment, our research team often connects the regional simulations of VIC-CropSyst to river system and water management models (e.g., ColSim and Yak-RW). These research efforts usually focus on understanding the dynamics between large-scale water supply, agricultural water demand, and the operation of dams and reservoirs. We provided more information about these applications in the manuscript.

Added/Modified:

*VIC-CropSyst has also been used in conjunction with reservoir models (e.g. ColSim; Wittwer et al., 2001 and YAK-RW; Zagona et al., 2001) to calculate the deficit irrigation fraction (e.g. Barik et al., 2017; Malek, et al., in preparation; Rajagopalan et al., in preparation). In general, the following six steps can be used to calculate and apply a deficit fraction: 1) VIC-CropSyst simulates the hydrologic states such as runoff and base flow as well as the irrigation water demand, 2) a routing model (i.e. Lohmann et al., 1998) is used to simulate streamflow, 3) simulated flow is bias corrected against observed flow, 4) a river system model is used to include operation of dams and reservoir and estimate water availability, 5) the availability of water is compared with demand, and 6)a deficit fraction is calculated and VIC-CropSyst is run to simulate the impacts of irrigation deficit on the hydrologic cycle and crop yields.*

**Page 10, last para: "As with other hydrological models, the VIC model needs to be calibrated : : :". This is only part of the story: State of the art is to calibrate AND validate in a split-sample approach. So, are the results shown in Figures 5-10 from the calibration or from the validation period? If, for example, the results in Figure 7 are from the calibration period, I would expect them to be good.**

Authors' response: We thank the reviewer for this observation. The authors agree that this section of the original manuscript was not written in a clear and understandable fashion. We will make sure that this part is more explicitly explained in our revised submission. To answer the reviewer's question, we did not calibrate the VIC-CropSyst at the flux tower sites in this study. We used calibrated parameters developed

in a separate study by Maurer et al. (2002). They calibrated a standalone version of the VIC model over the entire United States in 1/8$^{th}$ resolution. We selected the grid cells that overlap with our study sites.

Added/Modified:

*As with other hydrological models, the VIC model needs to be calibrated for optimized performance over a specific region. Table 3 shows VIC's key calibration parameters; more information on calibration parameters and methods can be found in past VIC studies (e.g. Elsner et al., 2010; Liang et al., 1994; Maurer et al., 2002). We used calibrated parameters determined by Maurer et al. (2002) for each flux tower station (the last two columns of Table 3).. We also tested the sensitivity of soil moisture content, crop growth, and irrigation demand and losses to different calibration parameters using the ranges available in Column 3 of Table 3 and differences were negligible..*

**In this paper the authors present a regional version of a coupled model system, the VIC hydrological model and the CropSyst crop model. The objective of the coupled system is a.o. to evaluate the potential impact of adaptation measures taken by farmers on basin scale hydrology.**

The authors would like to thank the reviewer # 2 for all the constructive comments. The following addresses the general and specific comments.

**General comments**

**The authors could elaborate a little more on the potential applications of this coupled model system, as they make not clear what is the added value of the coupled system versus the individual models.**

Potential application

The primary focus of VIC-CropSyst model was to combine in a tightly-integrated framework the strengths of an existing mechanistic large-scale hydrologic model with a mechanistic crop growth, phenology, and management model, with some potential applications being around adaptation, but other applications as well such as understanding the role that agricultural processes have in driving larger-scale water and energy cycles.  This model predominantly targets large river basins with significant agricultural activities. Also because VIC-CropSyst mechanistically simulates irrigation demand and losses, it can be used over regions with intensive irrigation (e.g. agricultural river basins of the western U.S). VIC-CropSyst can be applied at regional, continental or global scales and can provide the scientific community and policy makers with helpful information about the impact of management decisions and climatic factors on agricultural productivity, and water supply and demand.

VIC-CropSyst can also be used to understand impacts of agricultural management practices (e.g. switching to a new irrigation system or a new crop variety), under historic and future climate, on evapotranspiration and surface characteristics such as LAI, soil moisture and return flow from irrigated lands. The model is already being used within earth system models to serve two main purposes i) feed socioeconomic and river system management tools with water supply, yield and irrigation demand; and ii) improve boundary conditions of atmospheric models over agricultural areas.

Added/Modified

*VIC-CropSyst  is used in conjunction with reservoir operation models in the CRB, and accounting for the process of water rights curtailment under shortages in Washington State and farmer response to curtailment, to identify the indirect impacts on climate change on agricultural production though changes in water availability (Rajagopalan et al., in preparation). The current version of VIC-CropSyst (v2, as described herein) was also used in the most recent Columbia River Basin water supply and demand projection for the 2030s (Barik et al., 2017; Hall et al., 2017). These water supply and demand studies were submitted to the Washington State Legislature in the years of 2011 and 2016 and provide detailed information for each watershed in eastern Washington to the entire CRB as a whole. This information is being used by the Legislature for long-term water supply planning.*

*VIC-CropSyst has been used to investigate different scenarios for renegotiation of the Columbia River Treaty (Rushi et al. 2017). Existing modeling efforts to date have focused primarily on the impact that treaty renegotiation would have on flood risk, hydropower generation, and environmental flows (Cosens,*

*2010; Hamlet and Lettenmaier, 1999a); assessment of the impact of CRT changes on irrigated agriculture along the Columbia Mainstem is a knowledge gap. Rushi et al. (2017), therefore applied VIC-CropSyst linked to ColSim to simulate the complex impacts of climate change and the Columbia River Treaty on hydrology and agriculture in the river basin and concluded that climate change i) shifts water supply towards earlier in the season, ii) reduces flood risk in the upper CRB while increases frequency and magnitude of floods in the middle and lower parts of the basin, iii) shifts water demand towards earlier in the season in some locations with mixed effects on water rights curtailment risk, and iv) reduces hydropower generation. The authors found that the considered CRT scenarios can improve power generation and agricultural water demand while preventing floods in an altered climate.*

*VIC-CropSyst is an effective tool for studying the large-scale aggregated impacts of local management decisions and phenomena. For example, VIC-CropSyst was applied by Malek et al. (in review) who found that climate change-induced increases in evaporative (consumptive) losses from irrigation systems and decreases in non-evaporative irrigation losses (i.e., runoff and deep percolation) would lead to a decrease in reusable return flow, which would negatively affect basin-wide water availability and productivity.*

*VIC-CropSyst has also been used over the Yakima River basin (YRB) to evaluate the impacts of climate change on decisions related to investment in irrigation technology (Malek et al., 2016; in prep.). Economic damages of future more frequent droughts (Vano et al., 2010) are considered the main incentive to invest in more efficient irrigation technology (Berger and Troost, 2014). To analyze future changes in regional irrigation patterns, Malek et al. (in prep.) used VIC-CropSyst in conjunction with an economic model and the RiverWare model (Zagona et al., 2001). Figure 11 shows a result of this integration to simulate historical (1981-2006) drought frequency and severity, and the percentage of the YRB's perennial crop growers who are simulated to switch to more efficient irrigation systems to minimize the negative consequences of droughts during the two decades of 1990-2000 and 2050-2060. Also, any changes in agricultural activities (e.g., switching to a new irrigation system) directly impacts the hydrology of agricultural fields, thus changing return flow timing and magnitude and the availability of water for downstream users; these downstream consequences can also be simulated by this modeling platform. This is an example of how the human-land-climate nexus can be captured through a modeling framework that simulates large-scale hydrologic processes and regional water availability in a highly cultivated basin, while capturing the dynamics of farm-level irrigation decisions.*

**what is the added value of the coupled system versus the individual models.**

1-VIC:

VIC simulates one crop type and growth stages of that crop type is simulated through monthly prescribed LAIs, which means that VIC does not mechanistically simulate agricultural processes such as crop development, biomass production, the impact of water heat and nutrient stresses on crop growth, and many other details provided by CropSyst in the VIC-CropSyst coupled version. Also VIC does not mechanistically simulate irrigation losses and only includes one irrigation type (sprinkler). Lack of these processes makes any estimation of irrigation water demand, transpiration and crop growth questionable and can lead to inaccuracy in simulation of water and energy cycles over agricultural areas. VIC-CropSyst responds to these shortcomings as it is an implementation of a well-established mechanistic crop model that simulates agricultural processes in a sophisticated manner. In VIC-CropSyst crop growth is

controlled by environmental conditions such as radiation, water availability, temperature, nutrient and CO2 concentration.

2- CropSyst

CropSyst is a cropping system model that is able to simulate agricultural processes mechanistically. Although the primary purpose of this study was to improve simulation of land surface processes through adding a cropping system (CropSyst) to a widely used hydrologic model (VIC), simulation of agriculture processes in CropSyst can also benefit from this coupling. CropSyst has been already used to simulate local-scale hydrologic processes, but it has not been developed to simulate regional water and energy cycles. VIC has a more sophisticated and mechanistic way of handling regional hydrologic cycle. Many studies (e.g. Elsner et al., 2010; Hamlet and Lettenmaier, 1999; Maurer et al., 2002) have used VIC to simulate runoff, baseflow, soil moisture and cold season processes to eventually estimate availability of water for irrigation. Also, the stand-alone CropSyst does not have a mechanistic irrigation module. Therefore, we argue that the coupled model can improve the usefulness and applicability of CropSyst especially over irrigated areas.

Added/Modified

*We coupled the VIC version 4.1.2-e with CropSyst-v4.15, although the coupled model will be updated with new versions of VIC and CropSyst as they become available. In a spatially-explicit manner, VIC-CropSyst is able to capture a large variety of crop groups: 1- cereal grains (e.g. winter and spring wheat, corn, barley, oats, sorghum),  2- vegetables and melons (e.g. dill, radish, mint, broccoli, cauliflower, cabbage, carrot, onion, cucumber and pumpkins, watermelon), 3- fruits and nuts (e.g. plum, apricot, cherry, grape, walnut, pear, peaches, apples, blubbery, strawberry, cranberry), 4- root crops (e.g. potato, sugar beet), 5- leguminous crops (e.g. green and dry bean, lentil, chickpea, pea), 6- forages (e.g. pasture, alfalfa, hay, grass, clover, grass), and 7- oil seeds (e.g. soybean, mustard, sunflower).*

**- The authors claim that the coupled model system can be used to evaluate the impact of certain agriculture related adaptation measures over the region or river basin, but I was surprised to see that this impact is only modelled in one way. The way I understand the model from this manuscript, is that irrigation water is assumed to be always available, but the source of this irrigation water is not discussed. Unless water is always extracted from deep confined groundwater layers, there should be an effect of water withdrawals for irrigation on streamflow and water availability downstream. Since VIC explicitly calculates streamflow, I think it is a missed opportunity not to include this interaction, especially since irrigation withdrawals have been implemented before eg. by Haddeland et al. To my understanding there is no consideration of water shortage for irrigation.**

We would like to thank the reviewer for this observation. We added a new section to the main body of the paper to clarify this.

VIC-CropSyst is being used in a variety of projects following these steps (as documented by Malek et al., in preparation): 1) VIC-CropSyst simulates the hydrologic states such as runoff and base flow as well as the irrigation water demand, 2) a routing model (i.e. Lohmann et al., 1998) is used to simulate streamflow, 3) simulated flow is bias corrected against observed flow, 4) a river system model is used to include operation of dams and reservoir and estimate water availability, 5) availability of water is compared with demand, and 6) deficit fraction is calculated and VIC-CropSyst is run to simulate the deficit scenarios. Malek et al. (in preparation) discussed the implementation of VIC-CropSyst in these six steps as a part of the Agricultural Spatial Economic Analysis Platform (ASEAP) to investigate how farmers should invest on more efficient irrigation systems as climate changes.

Haddeland et al. (2006) used a similar process using a simple reservoir management module within the routing code of Lohmann et al. (1998). However, Haddeland et al. (2006) used the VIC model in isolation of a cropping system model (that captures all of the crop-specific characteristics and management that influence irrigation demand), and developers of VIC-CropSyst believe that a crop model is important for accurate simulation of irrigation demand. However, the following section has been added to the paper to clarify this issue:

Added/Modified

*Deficit irrigation*

*2.4.    Deficit irrigation*

[revised manuscript text omitted]

**- For sake of reproducibility, the authors should include more background of the models and equations used**

Authors' response: We appreciate this observation. More details on algorithms used in VIC-CropSyst and its irrigation module were added to address this weakness.

**Specific comments**

**- The abstract would benefit from a little more text on potential application of this model, and more specific on how it can be used to inform 'policy and best management practices to promote sustainable agriculture'. What can the model do, that cannot be done without a model?**

Authors' response: We would like to thank the reviewer; more information on the application of the model has been added to the abstract.

Added/Modified

*Because VIC-CropSyst combines two widely-used and mechanistic models (for crop growth phenology, growth, and management; and macroscale hydrology), it can provide realistic and hydrologically-consistent simulations of water availability, crop water requirement for irrigation, and agricultural productivity for both irrigated and dryland systems. This allows VIC-CropSyst to provide managers and decision makers with reliable information on regional water stresses and their impacts on food production. Additionally, VIC-CropSyst is being used in conjunction with socio-economic models, river system models and atmospheric models to simulate feedback processes between regional water availability, agricultural water management decisions, and land-atmospheric interactions.*

**- Pg 2, r 6.. there are unanswered questions.... what are the unanswered questions and how are you going to address them? - Pg2. r 6. The consequences of what kind of decisions are not understood? Can you give an example of a situation where that happened and where the use of this model could have helped?**

Authors' response:

Many of the current land surface models (e.g., VIC stand-alone) do not have a mechanistic way to simulate agricultural processes; many others use fixed seasonally variable parameters (e.g. LAI) to represent crop development or simulate the crop processes through simplified versions of crop models (Elliott et al., 2014). To the best of our knowledge, there is no other land surface model for which an

equally sophisticated cropping system model has been added. VIC-CropSyst is a state of the art tool that facilitates a better understanding of regional water supply and demand as well as agricultural productivity. Such a tool can also open new doors to the simulation of interactions among human, climate, hydrologic and agriculture factors over intensely cultivated areas.

Many types of crops have been systematically ignored in land surface models; crop types that sometime play a significant role in regional economy of agricultural regions. Reliable information on responses of different crop varieties has implication for agricultural decisions that can potentially impact regional water and energy cycles (e.g. what crop might be curtailed when there is a water shortage). In the VIC-CropSyst we simulate more than ninety types of crops allowing us to take heterogeneity among different crop types into consideration and more accurately answer questions regarding socioeconomic aspects, such as agricultural benefits of constructing a reservoir.

Malek et al. (in preparation) used the VIC-CropSyst to simulate how future climate alters overall irrigation efficiency as well as different loss terms (i.e. direct evaporation from irrigation systems, evaporation from water trapped by canopy, soil evaporation, runoff and deep percolation), and discussed the regional water availability implications of such changes. To the best of our knowledge, none of the current land surface models are able to mechanistically simulate irrigation processes to answer questions like this.

Also because of the lack of mechanistic approach to simulate irrigation losses it is difficult to accurately capture how change in irrigation technology impacts return flows (which is crucial in downstream water availability of many agricultural basins); this tool captures that.

Moreover, this tool (in conjunction with river system models such as ColSim and Yak-RW) is being used to capture unintended consequences of certain adaptation decisions. For example, when farmers switch to new irrigation technology to improve farm-level irrigation efficiency, they may also reduce valuable return flow that contributes to downstream water availability. Return flow reduction also impacts seasonality and magnitude of instream flow which directly affects ecosystems and hydroelectric generation.

Added/Modified

*Despite existing research on food scarcity, there are still unanswered questions about the relationship between food supply and the nexus of water resources, agriculture and human decisions. For example, how expectations of future climatic conditions influence farmer behaviour such as capital intensive switches in technology or cropping systems, is not well understood. Such scenarios require a simulation tool that can capture large-scale hydrologic processes while accurately simulating the impacts of climate, management, and water availability on different crop types. Moreover, regional consequences of decisions intended to mitigate the damages of future stressors are not well understood (Robertson and Swinton, 2005). For example, improvement in the efficiency of irrigation systems may increase consumptive water uses and lead to a reduction in return flow from irrigated areas (Causapé et al., 2004; Gosain et al., 2005). Return flow plays a significant role in the water availability of many agricultural regions; e.g., 40% of the water availability at the Yakima River's Parker Gauge in an average year is generated through return flows from upstream lands (USBR, 2010). Ecosystems and hydroelectric generation are also impacted as return flow changes. These knowledge gaps limit our ability to explore*

*viable adaptation strategies, particularly in understanding unintended consequences. Integrated modeling platforms can contribute to the systems-level understanding of dynamics between agricultural processes, large-scale water resources management decisions, and land-atmospheric interactions.*

**- P3 r 22. Can you give examples of the management decisions farmers can make?**

Authors' response: Table 6 in the manuscripts represents some examples of the management/adaptation decisions that can be handled by VIC-CropSyst. For example, this includes choosing among different irrigation systems (there are different irrigation types included in the model), or modifying the irrigation systems (e.g. using different sprinkler options available for different systems). There are also options to plant other crop types, such as new varieties with longer growing periods, an adaptation strategy that increases the opportunity for photosynthesis. Other decision variables include a choice to fallow or deficit-irrigate lands during drought. Other decisions that can be informed with future model developments are related to nutrient management, tillage practices, and rotation options.

Added/Modified

*While farmers can adjust their management decisions to reduce the negative impacts of climate change (e.g., switching to more-efficient irrigation technologies, more drought-tolerant crop types, varieties with longer growing periods, and precision agriculture), these human decisions can result in unintended impacts on regional water and energy cycles.*

**- P4 r1. What is meant by 'large scale results'?**

Authors' response: To clarify this, we modified the text as shown below.

Added/Modified

*Here, we define large-scale results as regionally-aggregated responses of agriculture to changes that can impact scales greater than a single cultivated field, such as a policy change (e.g., water law), climate-related impacts (e.g. warming-induced reductions in summer water availability), or development of large-scale infrastructure (e.g., a large reservoir).*

**- P5. I am surprised to see that you are not referring to other models that are also capable of relating hydrology to crop production.**

Authors' response: We have provided a more comprehensive literature review to address this and other related reviewer comments.

**- P5. Could you explain a little better why the (vertical) soil water balance of VIC is better than the one that was originally included in CropSyst? Since you are not using the lateral flow generated in VIC, the advantage of this coupling is not completely clear (to me).**

Authors' response:

We clarified this in the text as follows.

Added/Modified

*: In the integrated VIC-CropSyst model, CropSyst's soil hydrology is turned off, allowing VIC to simulate soil hydrologic processes, including the movement of water in soil, bare soil evaporation, and the generation of runoff and baseflow. We did this to retain consistency in all of the hydrologic processes. Standalone VIC and CropSyst use different soil hydrologic assumptions to simulate processes related to soil water movement and the generation of runoff and baseflow; these inconsistencies can lead to an inaccurate simulation of irrigation demand and crop productivity.*

**- P6. L14. It would be good to have a little more information on the crop model, since the information given here is very limited. E.g. which crops are included, how are sowing and harvest dates determined, is there any management included, how is yield calculated.**

Authors' response: we provided a more complete description of the CropSyst:

*In CropSyst the daily biomass production is restricted to the minimum of the two following biomass generation routines: i) radiation-based biomass production, and ii) transpiration-based biomass production. After simulation of potential biomass, CropSyst takes water, heat, freezing and nutrient stresses into account to calculate the actual yield. These stresses also modify other crop processes such as transpiration and LAI. Stress sensitivity varies during different phonological periods (e.g. from flowering to maturity). Root occurrence varies in each of the soil layers and depends on the root growth deeper into the soil during biomass development; thus, crop water and nutrient uptake also varies by soil layer. While the start and last date of the growing period is an input to the model, actual crop growth starts after a certain amount of thermal accumulation has been achieved during this user-specified growing period. Crop growth and development is also a function of thermal accumulation, affecting actual harvest date and other growth stages.*

**- P8. L6. As long as the paper describing the irrigation module in more detail is unpublished, it is difficult to judge the model, so a little more detail regarding algorithms is required here.**

Authors' response: Thanks for the comments we included more details to the irrigation section, specifically in how each of the loss terms are calculated.

Added/Modified

The following formulas are used to calculate $E_c$ and $E_d$ from sprinkler and center pivot irrigation systems. *Evaporation from Irrigation Intercepted Water ($E_c$): to calculate Ec, VIC-CropSyst uses the original VIC method (Liang et al., 1994). To avoid overestimation of Ec in agricultural areas, we used the equation developed by Kang et al., (2005) to set the maximum Ec. Evaporation from Irrigation Droplets ($E_d$). Users have the option to calculate Ed using one of the following two methods:*

*1- Malek et al., (2016, in prep.):*

$$E_d = ET_p \times \left(\frac{1}{D}\right)^{0.52} \times \left(\frac{V_0 \sin(\theta)}{g} + \frac{\sqrt{V_0^2 \sin^2(\theta) + 2g(Y_0 - Y)}}{g}\right)^{1.57} \tag{4}$$

*where $Y_0$(m) is height of nozzle; $Y$ (m) is canopy height; $V_0$ (m/sec) is initial velocity of the irrigation water which depends on Irrigation system pressure H (m), nuzzle coefficient $c_d$, and initial angle of sprinkler $\theta$; $A_p$ is irrigated area at a time; D (mm) is the droplet diameter and $ET_p$ (mm/$\Delta t$) is potential evapotranspiration.*

*2- Playán et al., (2005):*

*For sprinkler:* $\quad E_d = 20.3 + 0.214\, U^2 - 0.00229\, RH^2$ $\tag{5}$

*For moving laterals and center pivot:* $E_d = -2.1 + 1.91\, U^2 + 0.231\, T$ $\tag{6}$

*where T (C) is the air temperature; U ($m\ s^{-1}$) is wind speed; and RH (%) is the relative humidity.*

*Deep Percolation Loss (Dp)*

*Dp is defined as irrigated water which penetrates below the root zone. Therefore, after an irrigation event the amount of water that enters the base flow layer and becomes inaccessible for crop roots is considered a deep percolation loss.*

*Runoff Losses ($R_o$)*

*Ro depends on soil infiltration rate and irrigation intensity. Whenever irrigation intensity is higher than soil infiltration capacity, runoff is generated as follows,*

$$R_o = \frac{Ir}{t_{irr}} - f$$
*(7)*

*where f is the infiltration rate $\left(\frac{mm}{hr}\right)$, $I_r$ is the amount of irrigation water applied in each event ($mm$) and $t_{irr}$ is the duration of irrigation ($hr$). Although irrigation intensity is usually a management decision, soil texture and hydraulic conductivity are assumed to be the key considerations in a well-managed irrigation system; therefore in the beginning of simulation, VIC-CropSyst estimates the irrigation duration ($I_{du}$) using the soil characteristics of each gridcell. The calculated $I_{du}$ is used to estimate infiltration opportunity time of surface irrigation, rotation time in center pivot, and overlap and layout of sprinklers in solid-set, wheel move and big-gun irrigation systems. Then approximated irrigation intensity is compared with the irrigation infiltration rate ($f$). VIC-CropSyst uses the following equation developed by Philip, (1957) to estimate the infiltration rate,*

$$f = \frac{1}{2} S\, T_i^{-0.5} + K_s$$
*(8)*

*where $K_s \left(\frac{mm}{hr}\right)$ is the hydraulic conductivity and S is the sorptivity which is estimated through Rawls et al., (1992) formula and is calculated based on soil texture and initial water content. Therefore in VIC-CropSyst Ro depends on irrigation system, soil type, initial soil moisture as well as the intensity of water reaching to soil.  Details of the runoff calculations are presented by Malek, et al. (2016, in prep).*

**- P8. It is very impressive that the model is able to simulate over 40 different irrigation systems, but it would be good to briefly descibe how differences between those systems are implemented in the model and which assumptions are made.**

Authors' response: There are actually only four major categories with subcategories in each and the flexibility to adjust characteristics within these groups. This has been clarified and details added; please see below.

Added/Modified

*Currently, VIC-CropSyst simulates four major categories of irrigation systems: surface, center pivot, sprinkler, and drip. Each category includes subcategories. Drip systems include surface and subsurface drip irrigation. In surface drip irrigation, water is applied on the soil surface, while in subsurface drip irrigation, water is applied below the surface and will not lead to any soil evaporative losses. Surface irrigation includes furrow, rill, and border irrigation, and the main difference between these three systems is in their wetted surface area, which is smaller in a furrow system. Center pivots are represented by eighteen different types of sprinklers that fall into two subcategories: impact and spray sprinklers. Impact sprinklers generally have a greater discharge rate and wetted radius. Sprinkler systems in VIC-CropSyst include seventeen nozzles from three major subcategories: solid set, big gun, and moving wheels. The subcategories differ in terms of discharge, wetted diameter, height, droplet size, and other aspects. The characteristics of these systems have been collected from different scientific papers, reports, and commercial catalogs, including Nelson Co. (2014) and RainBird (2014). This level of detail offers a*

*more accurate representation of irrigation practices, and it will help users to simulate the adaptation of different irrigation and management scenarios.*

**- P8 l22. Which crops are included?**

Authors' response: Authors included the following table to clarify this:

Added/Modified

*In a spatially-explicit manner, VIC-CropSyst is able to capture a large variety of crop groups: 1- cereal grains (e.g. winter and spring wheat, corn, barley, oats, sorghum),  2- vegetables and melons (e.g. dill, radish, mint, broccoli, cauliflower,  cabbage, carrot, onion, cucumber and pumpkins, watermelon), 3- fruits and nuts (e.g. plum, apricot, cherry, grape, walnut, pear, peaches, apples, blubbery, strawberry, cranberry), 4- root crops (e.g. potato, sugar beet), 5- leguminous crops (e.g. green and dry bean, lentil, chickpea, pea), 6- forages (e.g. pasture, alfalfa, hay, grass, clover, grass), and 7- oil seeds (e.g. soybean, mustard, sunflower).*

**- P8 l29. For readibility, it would be good to write out the meaning of Esi in this sentence and put Esi between brackets.**

Authors' response: Authors edited wrote out the Esi and put it into a brackets

Added/Modified:

*In the drip and surface categories, evaporative losses happen only from the soil surface because irrigation happens below the canopy level. Irrigation takes place above the canopy in sprinkler and center pivot systems; therefore, evaporation from canopy-intercepted water ($E\_c$) and the direct loss from droplets ($E\_d$) are considered as major irrigation losses.  VIC-CropSyst neglects evaporative losses from soil ($E\_si$) for sprinkler and center pivot systems because energy is more readily available for water above the canopy and it suppresses the below-canopy evaporation (Uddin et al., 2013; Yonts et al., 2000).*

**- P9 l8. What is the equation, I think it would be good to add it here. - Idem for the equations in line 12 and line 14 (referring to an equation in an article in preparation is not ideal).**

Authors' response: All the equations have been included in the manuscript.

**- P10 l5. The simulated variables that are compared, could results been shown for all the mentioned variables?**

Authors' response:

Added/ Modified:

*VIC-CropSyst's simulated soil moisture, ET, yield and irrigation water demand were compared to observed data obtained from the FLUXNET network (Baldocchi et al., 2001). Simulated LAI was evaluated against Moderate Resolution Imaging Spectroradiometer (MODIS) remote sensing observations (Cohen et al., 2006). We also evaluated regional performance of VIC-CropSyst in simulation of ET over the U.S. Pacific Northwest, including the states of Washington, Idaho and Oregon. Other studies such as Malek et at (in preparation a and b), Rajagopalan, et al., (in preparation), Barik et al., (2017), Hall et al., 2017), Yorgey et al., (2011) evaluated VIC-CropSyst in its capability to capture regional irrigation demand, naturalized streamflow, observed flow, county level yield, snow water equivalent, and  irrigation efficiency.*

**- P10 l18. The soil files were modified using available information, could this be explained?**

Authors' response: we started from the Maurer et al (2002) soil file and we replaced its sand content with sand content available at the study site. We also added the clay percentages to the Maurer (2002)'s soil file.

Added/Modified:

*We replaced its sand content with data available at the study site. We also added the clay percentages to Maurer et al. (2002)'s soil file. In our simulation, VIC-CropSyst reads the sand and clay content and uses pedo-transfer functions developed by Saxton et al. (1986) to generate saturated hydraulic conductivity, bulk density, air entry potential, the b coefficient of Campbell, (1974)'s soil retention curve, field capacity, wilting point, and porosity.*

**- P11, l 19. Are those climate data the same as the climate data mentioned on p 10 (DAYMET)? This is somewhat confusing.**

Authors' response: We did not use the DAYMET data to run the model over three states of Washington, Oregon and Idaho. We used the data prepared by Abatzoglou and Brown, (2012). We added a table to clarify this in the text.

Added/Modified

*Table 5- Soil, climate, vegetation and crop information used for regional evaluation of VIC-CropSyst over the U.S. Pacific Northwest. The resolution of the input data was 1/16$^{th}$ °.*

| Input | Source | Information used by VIC-CropSyst |
|---|---|---|
| *Weather* | *Abatzoglou and Brown (2012)* | *precipitation, minimum and maximum temperature and wind speed* |
| *Soil* | *STATSGO (Schwarz and Alexander, 1995)* | *latitude, longitude, sand and clay content, hydraulic conductivity, field capacity, bulk density, etc.* |
| *Crop/Vegetation* | *USDA/WSDA vegetation distribution maps (Boryan et al., 2011; Yorgey et al., 2011)* | *crop type, acreage, irrigation systems, etc.* |

**- P 11 line 25. I understood from table 2 and different figures that simulations were made for corn only, how was the crop distribution information used?**

Authors' response: We evaluated VIC-CropSyst at point and regional scales. Over our point-scale sites (Nebraska and Illinois), we evaluated VIC-CropSyst for corn. However, over the region, we applied the model for all of the major crop types that are grown in the region.

Added/Modified:

*The performance of VIC-CropSyst was evaluated at both regional (over the U.S. Pacific Northwest) and point scales. Point-scale evaluation involved using two flux tower sites located in agricultural fields in the U.S. (Nebraska and Illinois).*

**- P12 section 3.3.1 Is water shortage for irrigation not considered at all? Could that be an issue in this region?**

Authors' response: Although irrigation shortage can be simulated by VIC-CropSyst (in combination with reservoir models like ColSim), we were not able to find any record of deficit irrigation management strategy at this specific site and did not apply deficit irrigation. Also, as figure 7 shows, there is no clear trend of yield overestimation by the model that strongly supports water shortage as an issue.

Added/Modified

*Although Figure 7 does not show a systematic overestimation by the model, a combination of inaccurate meteorological data, missing processes (e.g. lack of VPD feedback as discussed in section 3.1.2) and unrecorded conditions such as insufficient irrigation water or heat stress can contributes to these discrepancies.*

**- P12 section 3.1.2. Could the overestimation of etp also have to do with water shortage, the used crop parameterization?**

Authors' response: Because there was no mentioning of any deficit irrigation at Nebraska-irrigated site and also because a similar trend was observed at the non-irrigated site, we originally assumed that the corn crop is well-watered. However, we do agree that this can also be an uncertainty that can lead to discrepancies. Hence, we added this to the text as well.

Authors added the following to clarify this:

Added/Modified:

*Inaccuracy of the meteorological data or uncertainties related to unrecorded management practices such as deficit irrigation can be other sources of error.*

**- P12 3.1.3 could you describe a little better how yields are calculated and also reflect on why the variability for irrigated yields is not captured.**

There can be different reasons that the variability of the yield is not fully captured at the irrigated site: inaccuracy in climatic data, missing processes in the model such as the one discussed in section 3.1.2 (i.e. we do not simulate the feedback of irrigation evaporative losses on ambient temperature and VPD, which can change performance of stomata and rate of photosynthesis), and unrecorded heat and water stresses. However, because there was no systematic overestimation of the yield we could not conclude that a single unrecorded stress (e.g. water stress and heat stress) is responsible for this.

Authors believe that the discussion added to this section clarifies this:

Added/Modified:

*Although Figure 7 does not show a systematic overestimation by the model, a combination of inaccurate meteorological data, missing processes (e.g. lack of VPD feedback as discussed in section 3.1.2) and unrecorded conditions such as insufficient irrigation water or heat stress can contributes to these discrepancies..*

**- P13 3.1.4 It would be interesting to see some reflection on the meaning of errors.**

We added some texts to clarify this:

*The discrepancies may relate to the use of pedotransfer functions that convert soil textural characteristics to soil hydraulic properties (e.g. field capacity, permanent wilting point and hydraulic conductivity) for use in VIC-CropSyst (Pachepsky and Rawls, 1999; Tietje and Hennings, 1996). Also, scale discrepancies between the sensors' point-scale observation and the grid-scale simulation (Crow et al., 2012; Robinson et al., 2008) and inaccuracy of meteorological and soil data can be other sources of error. Additionally, imperfections in model processes such as soil water movement, evapotranspiration and irrigation loss calculation can contribute to the error.*

**- P14 I think it is good to emphasize here that the model can be used to evaluate the cumulative effects of large scale implementation of selected adaptation strategies over a basin or watershed.**

Authors added this:

*VIC-CropSyst can be used to investigate the hydrologic and atmospheric impacts of these adaptation decisions.*

This part in 4.2.1 can also help explaining the large scale consequences.

*Also, any changes in agricultural activities (e.g., switching to a new irrigation system) directly impacts the hydrology of agricultural fields, thus changing return flow timing and magnitude and the availability of water for downstream users; these downstream consequences can also be simulated by this modeling platform. This is an example of how the human-land-climate nexus can be captured through a modeling framework that simulates large-scale hydrologic processes and regional water availability in a highly cultivated basin, while capturing the dynamics of farm-level irrigation decisions.*

Reviewer # 1

**Major comments:**

**"VIC-CropSyst: A regional-scale modeling platform to simulate the nexus of climate, hydrology, cropping systems,**

5 **and human decisions" by Malek et al. In this paper the authors describe a simulation platform that captures the nexus of land, atmosphere, and human processes in one model. To this end, they have coupled the macroscale Variable Infiltration Capacity (VIC) hydrologic model and the CropSyst agricultural model. The paper is well written, good to understand and the results are well described. The topic of the study is of interest for scientists and natural resources decision makers.**

10 **However, there are some shortcomings, and the major in my view is that they ignore the huge amount of literature and work which already has been done in this direction. The authors present the topic of the study, to fully couple hydrological and agricultural models in one system considering feedbacks, as if this is an entirely new field. Cited are only global scale studies with comparable approaches but not so far developed. But the case studies given in this paper are at the regional and even local scale. And at the regional scale, first attempts to couple hydrological and**

15 **crop models started already in the late 70ties. A prominent example is SWAT (Soil and Water Assessment tool), nowadays also applied at the continental and global scale, and many other exist. At the global scale, the models ORCHIDEE and LPJmL have coupled water and crop modules etc.**

First and foremost, the authors would like to thank Reviewer #1 for this constructive feedback of the article.

The authors thank the reviewer for this insightful comment. The authors agree that the incorporation of agricultural processes

20 in hydrologic models such as SWAT dates back to the early stages of computer models, and this was not acknowledged in the manuscript. VIC and CropSyst are well-established large-scale hydrology and cropping systems models, respectively. The original intent was to present VIC-CropSyst as a contribution to large-scale land surface models due to its versatility and combined mechanistic simulation of crop and agricultural management processes as well as hydrologic processes. To address the reviewer's concern, in the revised manuscript, we incorporated a more comprehensive literature review that takes

25 various types of hydrologic models into account.

Added/Modified

[revised manuscript text omitted]

**Answer to minor comments:**

**Page 3, last para: How do you define return flow?**

In this study, we are referring to the USBR (2010) definition of return flow as a non-evaporative, reusable loss of water through conveyance systems and the field-level application of irrigation water. In the revised manuscript, we added some explanation to clarify this return flow definition.

Added/Modified:

*In many agricultural basins, the availability of water for downstream users depends greatly on the return flow from upstream lands, which mainly .comes from non-evaporative, reusable loss of water through conveyance systems and field-level application of irrigation water.*

**Page 4, second para: ": : :the current state of LSMs is not capable of capturing agricultural processes in a detailed manner". However, the literature cited is mostly older than 2010, and the most recent 2014. This is not the current state.**

Authors' response: In our new submission, we have done an up-to-date literature review and added more information about recently published works on this topic.

**Page 6, first para: Does VIC consider reservoirs and other water management measures?**

Authors' response: While VIC-CropSyst does not simulate reservoir directly nor the potential management decisions in operating these waterbody compartment, our research team often connects the regional simulations of VIC-CropSyst to river system and water management models (e.g., ColSim and Yak-RW). These research efforts usually focus on understanding the dynamics between large-scale water supply, agricultural water demand, and the operation of dams and reservoirs. We provided more information about these applications in the manuscript.

Added/Modified:

*VIC-CropSyst has also been used in conjunction with reservoir models (e.g. ColSim; Wittwer et al., 2001 and YAK-RW; Zagona et al., 2001) to calculate the deficit irrigation fraction (e.g. Barik et al., 2017; Malek, et al., in preparation; Rajagopalan et al., in preparation). In general, the following six steps can be used to calculate and apply a deficit fraction: 1) VIC-CropSyst simulates the hydrologic states such as runoff and base flow as well as the irrigation water demand, 2) a routing model (i.e. Lohmann et al., 1998) is used to simulate streamflow, 3) simulated flow is bias corrected against observed flow, 4) a river system model is used to include operation of dams and reservoir and estimate water availability, 5) the availability of water is compared with demand, and 6)a deficit fraction is calculated and VIC-CropSyst is run to simulate the impacts of irrigation deficit on the hydrologic cycle and crop yields.*

**Page 10, last para: "As with other hydrological models, the VIC model needs to be calibrated : : :". This is only part of the story: State of the art is to calibrate AND validate in a split-sample approach. So, are the results shown in Figures 5-10 from the calibration or from the validation period? If, for example, the results in Figure 7 are from the calibration period, I would expect them to be good.**

Authors' response: We thank the reviewer for this observation. The authors agree that this section of the original manuscript was not written in a clear and understandable fashion. We will make sure that this part is more explicitly explained in our revised submission. To answer the reviewer's question, we did not calibrate the VIC-CropSyst at the flux tower sites in this study. We used calibrated parameters developed in a separate study by Maurer et al. (2002). They calibrated a standalone version of the VIC model over the entire United States in $1/8^{th}$ resolution. We selected the grid cells that overlap with our study sites.

Added/Modified:

*As with other hydrological models, the VIC model needs to be calibrated for optimized performance over a specific region. Table 3 shows VIC's key calibration parameters; more information on calibration parameters and methods can be found in past VIC studies (e.g. Elsner et al., 2010; Liang et al., 1994; Maurer et al., 2002). We used calibrated parameters determined by Maurer et al. (2002) for each flux tower station (the last two columns of Table 3).. We also tested the sensitivity of soil moisture content, crop growth, and irrigation demand and losses to different calibration parameters using the ranges available in Column 3 of Table 3 and differences were negligible..*
* * *
**Reviewer # 2**

**In this paper the authors present a regional version of a coupled model system, the VIC hydrological model and the CropSyst crop model. The objective of the coupled system is a.o. to evaluate the potential impact of adaptation measures taken by farmers on basin scale hydrology.**

The authors would like to thank the reviewer # 2 for all the constructive comments. The following addresses the general and specific comments.

**General comments**

**The authors could elaborate a little more on the potential applications of this coupled model system, as they make not clear what is the added value of the coupled system versus the individual models.**

Potential application

The primary focus of VIC-CropSyst model was to combine in a tightly-integrated framework the strengths of an existing mechanistic large-scale hydrologic model with a mechanistic crop growth, phenology, and management model, with some potential applications being around adaptation, but other applications as well such as understanding the role that agricultural processes have in driving larger-scale water and energy cycles. This model predominantly targets large river basins with significant agricultural activities. Also because VIC-CropSyst mechanistically simulates irrigation demand and losses, it can be used over regions with intensive irrigation (e.g. agricultural river basins of the western U.S). VIC-CropSyst can be applied at regional, continental or global scales and can provide the scientific community and policy makers with helpful

information about the impact of management decisions and climatic factors on agricultural productivity, and water supply and demand.

VIC-CropSyst can also be used to understand impacts of agricultural management practices (e.g. switching to a new irrigation system or a new crop variety), under historic and future climate, on evapotranspiration and surface characteristics such as LAI, soil moisture and return flow from irrigated lands. The model is already being used within earth system models to serve two main purposes i) feed socioeconomic and river system management tools with water supply, yield and irrigation demand; and ii) improve boundary conditions of atmospheric models over agricultural areas.

Added/Modified

*VIC-CropSyst is used in conjunction with reservoir operation models in the CRB, and accounting for the process of water rights curtailment under shortages in Washington State and farmer response to curtailment, to identify the indirect impacts on climate change on agricultural production though changes in water availability (Rajagopalan et al., in preparation). The current version of VIC-CropSyst (v2, as described herein) was also used in the most recent Columbia River Basin water supply and demand projection for the 2030s (Barik et al., 2017; Hall et al., 2017). These water supply and demand studies were submitted to the Washington State Legislature in the years of 2011 and 2016 and provide detailed information for each watershed in eastern Washington to the entire CRB as a whole. This information is being used by the Legislature for long-term water supply planning.*

*VIC-CropSyst has been used to investigate different scenarios for renegotiation of the Columbia River Treaty (Rushi et al. 2017). Existing modeling efforts to date have focused primarily on the impact that treaty renegotiation would have on flood risk, hydropower generation, and environmental flows (Cosens, 2010; Hamlet and Lettenmaier, 1999a); assessment of the impact of CRT changes on irrigated agriculture along the Columbia Mainstem is a knowledge gap. Rushi et al. (2017), therefore applied VIC-CropSyst linked to ColSim to simulate the complex impacts of climate change and the Columbia River Treaty on hydrology and agriculture in the river basin and concluded that climate change i) shifts water supply towards earlier in the season, ii) reduces flood risk in the upper CRB while increases frequency and magnitude of floods in the middle and lower parts of the basin, iii) shifts water demand towards earlier in the season in some locations with mixed effects on water rights curtailment risk, and iv) reduces hydropower generation. The authors found that the considered CRT scenarios can improve power generation and agricultural water demand while preventing floods in an altered climate.*

*VIC-CropSyst is an effective tool for studying the large-scale aggregated impacts of local management decisions and phenomena. For example, VIC-CropSyst was applied by Malek et al. (in review) who found that climate change-induced increases in evaporative (consumptive) losses from irrigation systems and decreases in non-evaporative irrigation losses (i.e., runoff and deep percolation) would lead to a decrease in reusable return flow, which would negatively affect basin-wide water availability and productivity.*

*VIC-CropSyst has also been used over the Yakima River basin (YRB) to evaluate the impacts of climate change on decisions related to investment in irrigation technology (Malek et al., 2016; in prep.). Economic damages of future more frequent droughts (Vano et al., 2010) are considered the main incentive to invest in more efficient irrigation technology (Berger and*

*Troost, 2014). To analyze future changes in regional irrigation patterns, Malek et al. (in prep.) used VIC-CropSyst in conjunction with an economic model and the RiverWare model (Zagona et al., 2001). Figure 11 shows a result of this integration to simulate historical (1981-2006) drought frequency and severity, and the percentage of the YRB's perennial crop growers who are simulated to switch to more efficient irrigation systems to minimize the negative consequences of*

5 *droughts during the two decades of 1990-2000 and 2050-2060. Also, any changes in agricultural activities (e.g., switching to a new irrigation system) directly impacts the hydrology of agricultural fields, thus changing return flow timing and magnitude and the availability of water for downstream users; these downstream consequences can also be simulated by this modeling platform. This is an example of how the human-land-climate nexus can be captured through a modeling framework that simulates large-scale hydrologic processes and regional water availability in a highly cultivated basin, while capturing*

10 *the dynamics of farm-level irrigation decisions.*

**what is the added value of the coupled system versus the individual models.**

1-VIC:

VIC simulates one crop type and growth stages of that crop type is simulated through monthly prescribed LAIs, which means that VIC does not mechanistically simulate agricultural processes such as crop development, biomass production, the

15 impact of water heat and nutrient stresses on crop growth, and many other details provided by CropSyst in the VIC-CropSyst coupled version. Also VIC does not mechanistically simulate irrigation losses and only includes one irrigation type (sprinkler). Lack of these processes makes any estimation of irrigation water demand, transpiration and crop growth questionable and can lead to inaccuracy in simulation of water and energy cycles over agricultural areas. VIC-CropSyst responds to these shortcomings as it is an implementation of a well-established mechanistic crop model that simulates

20 agricultural processes in a sophisticated manner. In VIC-CropSyst crop growth is controlled by environmental conditions such as radiation, water availability, temperature, nutrient and $CO_2$ concentration.

2- CropSyst

CropSyst is a cropping system model that is able to simulate agricultural processes mechanistically. Although the primary purpose of this study was to improve simulation of land surface processes through adding a cropping system (CropSyst) to a

25 widely used hydrologic model (VIC), simulation of agriculture processes in CropSyst can also benefit from this coupling. CropSyst has been already used to simulate local-scale hydrologic processes, but it has not been developed to simulate regional water and energy cycles. VIC has a more sophisticated and mechanistic way of handling regional hydrologic cycle. Many studies (e.g. Elsner et al., 2010; Hamlet and Lettenmaier, 1999; Maurer et al., 2002) have used VIC to simulate runoff, baseflow, soil moisture and cold season processes to eventually estimate availability of water for irrigation. Also, the stand-

30 alone CropSyst does not have a mechanistic irrigation module. Therefore, we argue that the coupled model can improve the usefulness and applicability of CropSyst especially over irrigated areas.

Added/Modified

*We coupled the VIC version 4.1.2-e with CropSyst-v4.15, although the coupled model will be updated with new versions of VIC and CropSyst as they become available. In a spatially-explicit manner, VIC-CropSyst is able to capture a large variety of crop groups: 1- cereal grains (e.g. winter and spring wheat, corn, barley, oats, sorghum), 2- vegetables and melons (e.g. dill, radish, mint, broccoli, cauliflower, cabbage, carrot, onion, cucumber and pumpkins, watermelon), 3- fruits and nuts (e.g. plum, apricot, cherry, grape, walnut, pear, peaches, apples, blubbery, strawberry, cranberry), 4- root crops (e.g. potato, sugar beet), 5- leguminous crops (e.g. green and dry bean, lentil, chickpea, pea), 6- forages (e.g. pasture, alfalfa, hay, grass, clover, grass), and 7- oil seeds (e.g. soybean, mustard, sunflower).*

**- The authors claim that the coupled model system can be used to evaluate the impact of certain agriculture related adaptation measures over the region or river basin, but I was surprised to see that this impact is only modelled in one way. The way I understand the model from this manuscript, is that irrigation water is assumed to be always available, but the source of this irrigation water is not discussed. Unless water is always extracted from deep confined groundwater layers, there should be an effect of water withdrawals for irrigation on streamflow and water availability downstream. Since VIC explicitly calculates streamflow, I think it is a missed opportunity not to include this interaction, especially since irrigation withdrawals have been implemented before eg. by Haddeland et al. To my understanding there is no consideration of water shortage for irrigation.**

We would like to thank the reviewer for this observation. We added a new section to the main body of the paper to clarify this.

VIC-CropSyst is being used in a variety of projects following these steps (as documented by Malek et al., in preparation): 1) VIC-CropSyst simulates the hydrologic states such as runoff and base flow as well as the irrigation water demand, 2) a routing model (i.e. Lohmann et al., 1998) is used to simulate streamflow, 3) simulated flow is bias corrected against observed flow, 4) a river system model is used to include operation of dams and reservoir and estimate water availability, 5) availability of water is compared with demand, and 6) deficit fraction is calculated and VIC-CropSyst is run to simulate the deficit scenarios. Malek et al. (in preparation) discussed the implementation of VIC-CropSyst in these six steps as a part of the Agricultural Spatial Economic Analysis Platform (ASEAP) to investigate how farmers should invest on more efficient irrigation systems as climate changes.

Haddeland et al. (2006) used a similar process using a simple reservoir management module within the routing code of Lohmann et al. (1998). However, Haddeland et al. (2006) used the VIC model in isolation of a cropping system model (that captures all of the crop-specific characteristics and management that influence irrigation demand), and developers of VIC-CropSyst believe that a crop model is important for accurate simulation of irrigation demand. However, the following section has been added to the paper to clarify this issue:

Added/Modified

*Deficit irrigation*

*2.4.     Deficit irrigation*

*VIC-CropSyst's deficit irrigation module requires two main inputs: a) a first approximation to the irrigation water demand obtained by generating time series of irrigation under no water stress condition using VIC-CropSyst, and b) deficit fractions that indicate the water availability. VIC-CropSyst then reads the amount of recorded irrigation from step one and applies the deficit fraction to simulate the agricultural and hydrologic processes under realistic water deficit conditions. The deficit*

5 *fraction can be either homogenously applied across the entire basin or separately specified for each farmer depending on water rights or other considerations. Also, VIC-CropSyst can apply the deficit fraction during different times of the year. For example, if the water deficit happens later in the season, VIC-CropSyst can adjust irrigation amounts according to the timing of water shortage.*

*VIC-CropSyst has also been used in conjunction with reservoir models (e.g. ColSim; Wittwer et al., 2001 and YAK-RW;*

10 *Zagona et al., 2001) to calculate the deficit irrigation fraction (e.g. Barik et al., 2017; Malek, et al., in preparation; Rajagopalan et al., in preparation). In general, the following six steps can be used to calculate and apply a deficit fraction: 1) VIC-CropSyst simulates the hydrologic states such as runoff and base flow as well as the irrigation water demand, 2) a routing model (i.e. Lohmann et al., 1998) is used to simulate streamflow, 3) simulated flow is bias corrected against observed flow, 4) a river system model is used to include operation of dams and reservoir and estimate water availability, 5)*

15 *the availability of water is compared with demand, and 6)a deficit fraction is calculated and VIC-CropSyst is run to simulate the impacts of irrigation deficit on the hydrologic cycle and crop yields.*

**- I miss the broader embedding of this research in the existing body of knowledge. This model is certainly not the first to combine a hydrology and crop model (eg. LPJmL), but the authors seem to mainly relate to their own research in the introduction.**

20 Authors' response: we added the following to strengthen our background section:

Added/Modified

[revised manuscript text omitted]

30  **- For sake of reproducibility, the authors should include more background of the models and equations used**

Authors' response: We appreciate this observation. More details on algorithms used in VIC-CropSyst and its irrigation module were added to address this weakness.

**Specific comments**

**- The abstract would benefit from a little more text on potential application of this model, and more specific on how it can be used to inform 'policy and best management practices to promote sustainable agriculture'. What can the model do, that cannot be done without a model?**

Authors' response: We would like to thank the reviewer; more information on the application of the model has been added to the abstract.

Added/Modified

*Because VIC-CropSyst combines two widely-used and mechanistic models (for crop growth phenology, growth, and management; and macroscale hydrology), it can provide realistic and hydrologically-consistent simulations of water availability, crop water requirement for irrigation, and agricultural productivity for both irrigated and dryland systems. This allows VIC-CropSyst to provide managers and decision makers with reliable information on regional water stresses and their impacts on food production. Additionally, VIC-CropSyst is being used in conjunction with socio-economic models, river system models and atmospheric models to simulate feedback processes between regional water availability, agricultural water management decisions, and land-atmospheric interactions.*

**- Pg 2, r 6.. there are unanswered questions.... what are the unanswered questions and how are you going to address them? - Pg2. r 6. The consequences of what kind of decisions are not understood? Can you give an example of a situation where that happened and where the use of this model could have helped?**

Authors' response:

Many of the current land surface models (e.g., VIC stand-alone) do not have a mechanistic way to simulate agricultural processes; many others use fixed seasonally variable parameters (e.g. LAI) to represent crop development or simulate the crop processes through simplified versions of crop models (Elliott et al., 2014). To the best of our knowledge, there is no other land surface model for which an equally sophisticated cropping system model has been added. VIC-CropSyst is a state of the art tool that facilitates a better understanding of regional water supply and demand as well as agricultural productivity. Such a tool can also open new doors to the simulation of interactions among human, climate, hydrologic and agriculture factors over intensely cultivated areas.

Many types of crops have been systematically ignored in land surface models; crop types that sometime play a significant role in regional economy of agricultural regions. Reliable information on responses of different crop varieties has implication for agricultural decisions that can potentially impact regional water and energy cycles (e.g. what crop might be curtailed when there is a water shortage). In the VIC-CropSyst we simulate more than ninety types of crops allowing us to take heterogeneity among different crop types into consideration and more accurately answer questions regarding socioeconomic aspects, such as agricultural benefits of constructing a reservoir.

Malek et al. (in preparation) used the VIC-CropSyst to simulate how future climate alters overall irrigation efficiency as well as different loss terms (i.e. direct evaporation from irrigation systems, evaporation from water trapped by canopy, soil evaporation, runoff and deep percolation), and discussed the regional water availability implications of such changes. To the

best of our knowledge, none of the current land surface models are able to mechanistically simulate irrigation processes to answer questions like this.

Also because of the lack of mechanistic approach to simulate irrigation losses it is difficult to accurately capture how change in irrigation technology impacts return flows (which is crucial in downstream water availability of many agricultural basins); this tool captures that.

Moreover, this tool (in conjunction with river system models such as ColSim and Yak-RW) is being used to capture unintended consequences of certain adaptation decisions. For example, when farmers switch to new irrigation technology to improve farm-level irrigation efficiency, they may also reduce valuable return flow that contributes to downstream water availability. Return flow reduction also impacts seasonality and magnitude of instream flow which directly affects ecosystems and hydroelectric generation.

Added/Modified

*Despite existing research on food scarcity, there are still unanswered questions about the relationship between food supply and the nexus of water resources, agriculture and human decisions. For example, how expectations of future climatic conditions influence farmer behaviour such as capital intensive switches in technology or cropping systems, is not well understood. Such scenarios require a simulation tool that can capture large-scale hydrologic processes while accurately simulating the impacts of climate, management, and water availability on different crop types. Moreover, regional consequences of decisions intended to mitigate the damages of future stressors are not well understood (Robertson and Swinton, 2005). For example, improvement in the efficiency of irrigation systems may increase consumptive water uses and lead to a reduction in return flow from irrigated areas (Causapé et al., 2004; Gosain et al., 2005). Return flow plays a significant role in the water availability of many agricultural regions; e.g., 40% of the water availability at the Yakima River's Parker Gauge in an average year is generated through return flows from upstream lands (USBR, 2010). Ecosystems and hydroelectric generation are also impacted as return flow changes. These knowledge gaps limit our ability to explore viable adaptation strategies, particularly in understanding unintended consequences. Integrated modeling platforms can contribute to the systems-level understanding of dynamics between agricultural processes, large-scale water resources management decisions, and land-atmospheric interactions.*

**- P3 r 22. Can you give examples of the management decisions farmers can make?**

Authors' response: Table 6 in the manuscripts represents some examples of the management/adaptation decisions that can be handled by VIC-CropSyst. For example, this includes choosing among different irrigation systems (there are different irrigation types included in the model), or modifying the irrigation systems (e.g. using different sprinkler options available for different systems). There are also options to plant other crop types, such as new varieties with longer growing periods, an adaptation strategy that increases the opportunity for photosynthesis. Other decision variables include a choice to fallow or deficit-irrigate lands during drought. Other decisions that can be informed with future model developments are related to nutrient management, tillage practices, and rotation options.

Added/Modified

*While farmers can adjust their management decisions to reduce the negative impacts of climate change (e.g., switching to more-efficient irrigation technologies, more drought-tolerant crop types, varieties with longer growing periods, and*
5  *precision agriculture), these human decisions can result in unintended impacts on regional water and energy cycles.*

**- P4 r1. What is meant by 'large scale results'?**

Authors' response: To clarify this, we modified the text as shown below.

Added/Modified

10  *Here, we define large-scale results as regionally-aggregated responses of agriculture to changes that can impact scales greater than a single cultivated field, such as a policy change (e.g., water law), climate-related impacts (e.g. warming-induced reductions in summer water availability), or development of large-scale infrastructure (e.g., a large reservoir).*

**- P5. I am surprised to see that you are not referring to other models that are also capable of relating hydrology to crop production.**

15  Authors' response: We have provided a more comprehensive literature review to address this and other related reviewer comments.

**- P5. Could you explain a little better why the (vertical) soil water balance of VIC is better than the one that was originally included in CropSyst? Since you are not using the lateral flow generated in VIC, the advantage of this**
20  **coupling is not completely clear (to me).**

Authors' response:

We clarified this in the text as follows.

Added/Modified

25  *: In the integrated VIC-CropSyst model, CropSyst's soil hydrology is turned off, allowing VIC to simulate soil hydrologic processes, including the movement of water in soil, bare soil evaporation, and the generation of runoff and baseflow. We did this to retain consistency in all of the hydrologic processes. Standalone VIC and CropSyst use different soil hydrologic assumptions to simulate processes related to soil water movement and the generation of runoff and baseflow; these inconsistencies can lead to an inaccurate simulation of irrigation demand and crop productivity.*

30  **- P6. L14. It would be good to have a little more information on the crop model, since the information given here is very limited. E.g. which crops are included, how are sowing and harvest dates determined, is there any management included, how is yield calculated.**

Authors' response: we provided a more complete description of the CropSyst:

*In CropSyst the daily biomass production is restricted to the minimum of the two following biomass generation routines: i) radiation-based biomass production, and ii) transpiration-based biomass production. After simulation of potential biomass, CropSyst takes water, heat, freezing and nutrient stresses into account to calculate the actual yield. These stresses also modify other crop processes such as transpiration and LAI. Stress sensitivity varies during different phonological periods*
5    *(e.g. from flowering to maturity). Root occurrence varies in each of the soil layers and depends on the root growth deeper into the soil during biomass development; thus, crop water and nutrient uptake also varies by soil layer. While the start and last date of the growing period is an input to the model, actual crop growth starts after a certain amount of thermal accumulation has been achieved during this user-specified growing period. Crop growth and development is also a function of thermal accumulation, affecting actual harvest date and other growth stages.*

**- P8. L6. As long as the paper describing the irrigation module in more detail is unpublished, it is difficult to judge the model, so a little more detail regarding algorithms is required here.**

Authors' response: Thanks for the comments we included more details to the irrigation section, specifically in how each of the loss terms are calculated.

15    Added/Modified

*The following formulas are used to calculate $E_c$ and $E_d$ from sprinkler and center pivot irrigation systems. Evaporation from Irrigation Intercepted Water ($E_c$): to calculate Ec, VIC-CropSyst uses the original VIC method (Liang et al., 1994). To avoid overestimation of Ec in agricultural areas, we used the equation developed by Kang et al., (2005) to set the maximum Ec. Evaporation from Irrigation Droplets ($E_d$). Users have the option to calculate Ed using one of the following two methods:*

20    *1- Malek et al., (2016, in prep.):*

$$E_d = ET_p \times \left(\frac{1}{D}\right)^{0.52} \times \left(\frac{V_0 \sin(\theta)}{g} + \frac{\sqrt{V_0^2 \sin^2(\theta) + 2g(Y_0 - Y)}}{g}\right)^{1.57} \qquad (4)$$

*where $Y_0(m)$ is height of nozzle; $Y$ (m) is canopy height; $V_0$ (m/sec) is initial velocity of the irrigation water which depends on Irrigation system pressure H (m), nuzzle coefficient $c_d$, and initial angle of sprinkler $\theta$; $A_p$ is irrigated area at a time; D (mm) is the droplet diameter and $ET_p$ (mm/$\Delta t$) is potential evapotranspiration.*

25    *2- Playán et al., (2005):*

*For sprinkler:*    $E_d = 20.3 + 0.214 \, U^2 - 0.00229 \, RH^2$         (5)

*For moving laterals and center pivot:*  $E_d = -2.1 + 1.91 \, U^2 + 0.231 \, T$     (6)

*where T (C) is the air temperature; U (m $s^{-1}$) is wind speed; and RH (%) is the relative humidity.*

*Deep Percolation Loss (Dp)*

*Dp is defined as irrigated water which penetrates below the root zone. Therefore, after an irrigation event the amount of water that enters the base flow layer and becomes inaccessible for crop roots is considered a deep percolation loss.*

*Runoff Losses ($R_o$)*

5 *$R_o$ depends on soil infiltration rate and irrigation intensity. Whenever irrigation intensity is higher than soil infiltration capacity, runoff is generated as follows,*

$$R_o = \frac{I_r}{t_{irr}} - f \qquad\qquad\qquad (7)$$

10 *where f is the infiltration rate $\left(\frac{mm}{hr}\right)$, $I_r$ is the amount of irrigation water applied in each event ($mm$) and $t_{irr}$ is the duration of irrigation ($hr$). Although irrigation intensity is usually a management decision, soil texture and hydraulic conductivity are assumed to be the key considerations in a well-managed irrigation system; therefore in the beginning of simulation, VIC-CropSyst estimates the irrigation duration ($I_{du}$) using the soil characteristics of each gridcell. The calculated $I_{du}$ is used to estimate infiltration opportunity time of surface irrigation, rotation time in center pivot, and overlap and layout of sprinklers*
15 *in solid-set, wheel move and big-gun irrigation systems. Then approximated irrigation intensity is compared with the irrigation infiltration rate (f). VIC-CropSyst uses the following equation developed by Philip, (1957) to estimate the infiltration rate,*

$$f = \frac{1}{2}S\,T_i^{-0.5} + K_s \qquad\qquad\qquad (8)$$

*where $K_s$ $\left(\frac{mm}{hr}\right)$ is the hydraulic conductivity and S is the sorptivity which is estimated through Rawls et al., (1992) formula and is calculated based on soil texture and initial water content. Therefore in VIC-CropSyst Ro depends on irrigation system, soil type, initial soil moisture as well as the intensity of water reaching to soil. Details of the runoff calculations are presented by Malek, et al. (2016, in prep).*

**- P8. It is very impressive that the model is able to simulate over 40 different irrigation systems, but it would be good to briefly descibe how differences between those systems are implemented in the model and which assumptions are made.**

Authors' response: There are actually only four major categories with subcategories in each and the flexibility to adjust
30 characteristics within these groups. This has been clarified and details added; please see below.

Added/Modified

*Currently, VIC-CropSyst simulates four major categories of irrigation systems: surface, center pivot, sprinkler, and drip. Each category includes subcategories. Drip systems include surface and subsurface drip irrigation. In surface drip irrigation, water is applied on the soil surface, while in subsurface drip irrigation, water is applied below the surface and will not lead to any soil evaporative losses. Surface irrigation includes furrow, rill, and border irrigation, and the main difference between these three systems is in their wetted surface area, which is smaller in a furrow system. Center pivots are represented by eighteen different types of sprinklers that fall into two subcategories: impact and spray sprinklers. Impact sprinklers generally have a greater discharge rate and wetted radius. Sprinkler systems in VIC-CropSyst include seventeen nozzles from three major subcategories: solid set, big gun, and moving wheels. The subcategories differ in terms of discharge, wetted diameter, height, droplet size, and other aspects. The characteristics of these systems have been collected from different scientific papers, reports, and commercial catalogs, including Nelson Co. (2014) and RainBird (2014). This level of detail offers a more accurate representation of irrigation practices, and it will help users to simulate the adaptation of different irrigation and management scenarios.*

**- P8 l22. Which crops are included?**

Authors' response: Authors included the following table to clarify this:

Added/Modified

*In a spatially-explicit manner, VIC-CropSyst is able to capture a large variety of crop groups: 1- cereal grains (e.g. winter and spring wheat, corn, barley, oats, sorghum), 2- vegetables and melons (e.g. dill, radish, mint, broccoli, cauliflower, cabbage, carrot, onion, cucumber and pumpkins, watermelon), 3- fruits and nuts (e.g. plum, apricot, cherry, grape, walnut, pear, peaches, apples, blubbery, strawberry, cranberry), 4- root crops (e.g. potato, sugar beet), 5- leguminous crops (e.g. green and dry bean, lentil, chickpea, pea), 6- forages (e.g. pasture, alfalfa, hay, grass, clover, grass), and 7- oil seeds (e.g. soybean, mustard, sunflower).*

**- P8 l29. For readability, it would be good to write out the meaning of Esi in this sentence and put Esi between brackets.**

Authors' response: Authors edited wrote out the Esi and put it into a brackets

Added/Modified:

*In the drip and surface categories, evaporative losses happen only from the soil surface because irrigation happens below the canopy level. Irrigation takes place above the canopy in sprinkler and center pivot systems; therefore, evaporation from canopy-intercepted water ($E_c$) and the direct loss from droplets ($E_d$) are considered as major irrigation losses. VIC-CropSyst neglects evaporative losses from soil ($E_{si}$) for sprinkler and center pivot systems because energy is more readily available for water above the canopy and it suppresses the below-canopy evaporation (Uddin et al., 2013; Yonts et al., 2000).*

**- P9 l8. What is the equation, I think it would be good to add it here. - Idem for the equations in line 12 and line 14 (referring to an equation in an article in preparation is not ideal).**

Authors' response: All the equations have been included in the manuscript.

**- P10 l5. The simulated variables that are compared, could results been shown for all the mentioned variables?**

5 Authors' response:

Added/ Modified:

*VIC-CropSyst's simulated soil moisture, ET, yield and irrigation water demand were compared to observed data obtained from the FLUXNET network (Baldocchi et al., 2001). Simulated LAI was evaluated against Moderate Resolution Imaging Spectroradiometer (MODIS) remote sensing observations (Cohen et al., 2006). We also evaluated regional performance of*
10 *VIC-CropSyst in simulation of ET over the U.S. Pacific Northwest, including the states of Washington, Idaho and Oregon. Other studies such as Malek et at (in preparation a and b), Rajagopalan, et al., (in preparation), Barik et al., (2017), Hall et al., 2017), Yorgey et al., (2011) evaluated VIC-CropSyst in its capability to capture regional irrigation demand, naturalized streamflow, observed flow, county level yield, snow water equivalent, and irrigation efficiency.*

**- P10 l18. The soil files were modified using available information, could this be explained?**

Authors' response: we started from the Maurer et al (2002) soil file and we replaced its sand content with sand content available at the study site. We also added the clay percentages to the Maurer (2002)'s soil file.

20 Added/Modified:

*We replaced its sand content with data available at the study site. We also added the clay percentages to Maurer et al. (2002)'s soil file. In our simulation, VIC-CropSyst reads the sand and clay content and uses pedo-transfer functions developed by Saxton et al. (1986) to generate saturated hydraulic conductivity, bulk density, air entry potential, the b coefficient of Campbell, (1974)'s soil retention curve, field capacity, wilting point, and porosity.*

25 **- P11, l 19. Are those climate data the same as the climate data mentioned on p 10 (DAYMET)? This is somewhat confusing.**

Authors' response: We did not use the DAYMET data to run the model over three states of Washington, Oregon and Idaho. We used the data prepared by Abatzoglou and Brown, (2012). We added a table to clarify this in the text.

Added/Modified

30 *Table 5- Soil, climate, vegetation and crop information used for regional evaluation of VIC-CropSyst over the U.S. Pacific Northwest. The resolution of the input data was $1/16^{th}°$.*

| *Input* | *Source* | *Information used by VIC-CropSyst* |
|---------|----------|-----------------------------------|
| *Weather* | *Abatzoglou and Brown (2012)* | *precipitation, minimum and maximum temperature and wind speed* |
| *Soil* | *STATSGO (Schwarz and Alexander, 1995)* | *latitude, longitude, sand and clay content, hydraulic conductivity, field capacity, bulk* |

| | | |
|---|---|---|
| | | density, etc. |
| Crop/Vegetation | USDA/WSDA vegetation distribution maps (Boryan et al., 2011; Yorgey et al., 2011) | crop type, acreage, irrigation systems, etc. |

**- P 11 line 25. I understood from table 2 and different figures that simulations were made for corn only, how was the crop distribution information used?**

Authors' response: We evaluated VIC-CropSyst at point and regional scales. Over our point-scale sites (Nebraska and Illinois), we evaluated VIC-CropSyst for corn. However, over the region, we applied the model for all of the major crop types that are grown in the region.

Added/Modified:

*The performance of VIC-CropSyst was evaluated at both regional (over the U.S. Pacific Northwest) and point scales. Point-scale evaluation involved using two flux tower sites located in agricultural fields in the U.S. (Nebraska and Illinois).*

**- P12 section 3.3.1 Is water shortage for irrigation not considered at all? Could that be an issue in this region?**

Authors' response: Although irrigation shortage can be simulated by VIC-CropSyst (in combination with reservoir models like ColSim), we were not able to find any record of deficit irrigation management strategy at this specific site and did not apply deficit irrigation. Also, as figure 7 shows, there is no clear trend of yield overestimation by the model that strongly supports water shortage as an issue.

Added/Modified

*Although Figure 7 does not show a systematic overestimation by the model, a combination of inaccurate meteorological data, missing processes (e.g. lack of VPD feedback as discussed in section 3.1.2) and unrecorded conditions such as insufficient irrigation water or heat stress can contributes to these discrepancies.*

**- P12 section 3.1.2. Could the overestimation of etp also have to do with water shortage, the used crop parameterization?**

Authors' response: Because there was no mentioning of any deficit irrigation at Nebraska-irrigated site and also because a similar trend was observed at the non-irrigated site, we originally assumed that the corn crop is well-watered. However, we do agree that this can also be an uncertainty that can lead to discrepancies. Hence, we added this to the text as well.

Authors added the following to clarify this:

Added/Modified:

*Inaccuracy of the meteorological data or uncertainties related to unrecorded management practices such as deficit irrigation can be other sources of error.*

**- P12 3.1.3 could you describe a little better how yields are calculated and also reflect on why the variability for irrigated yields is not captured.**

There can be different reasons that the variability of the yield is not fully captured at the irrigated site: inaccuracy in climatic data, missing processes in the model such as the one discussed in section 3.1.2 (i.e. we do not simulate the feedback of irrigation evaporative losses on ambient temperature and VPD, which can change performance of stomata and rate of photosynthesis), and unrecorded heat and water stresses. However, because there was no systematic overestimation of the yield we could not conclude that a single unrecorded stress (e.g. water stress and heat stress) is responsible for this.

Authors believe that the discussion added to this section clarifies this:

Added/Modified:

*Although Figure 7 does not show a systematic overestimation by the model, a combination of inaccurate meteorological data, missing processes (e.g. lack of VPD feedback as discussed in section 3.1.2) and unrecorded conditions such as insufficient irrigation water or heat stress can contributes to these discrepancies..*

**- P13 3.1.4 It would be interesting to see some reflection on the meaning of errors.**

We added some texts to clarify this:

*The discrepancies may relate to the use of pedotransfer functions that convert soil textural characteristics to soil hydraulic properties (e.g. field capacity, permanent wilting point and hydraulic conductivity) for use in VIC-CropSyst (Pachepsky and Rawls, 1999; Tietje and Hennings, 1996). Also, scale discrepancies between the sensors' point-scale observation and the grid-scale simulation (Crow et al., 2012; Robinson et al., 2008) and inaccuracy of meteorological and soil data can be other sources of error. Additionally, imperfections in model processes such as soil water movement, evapotranspiration and irrigation loss calculation can contribute to the error.*

**- P14 I think it is good to emphasize here that the model can be used to evaluate the cumulative effects of large scale implementation of selected adaptation strategies over a basin or watershed.**

Authors added this:

[revised manuscript text omitted]

---

## Author Response (AR2)

**Responses to the Editor, Dr. Müller**

**Dear Dr. Malek and co-authors,**

**I'm happy to now have 2 reviews for your revised paper (one of the original ones and a new reviewer). I apologize for the long turnaround time this revision needed.**
**However, both referees are now largely content with your revisions and only referee #3 raises 2 last points. On top of these points, I have some more minor points, all with respect to your discussion of other approaches.**
**I do see that referee #3's first point may be somewhat difficult to address, as deciding which model to use under which circumstances is often hypothetical and would require some testing and model (setup) evaluation. So maybe you can discuss this with some specific examples where some models may be fit for purpose by addressing relevant processes and others may not as they don't.**

We would like to thank the topical editor for these careful review and insightful comments which have improved the quality of the paper.

Specific comments:
**As I happen to know the gridded crop model development to some extent, and LPJmL in particular, I noticed that you list JULES and ORCHIDEE as simplified model, which is not always true, see**
**\* Wang et al. 2016 http://www.geosci-model-dev.net/9/857/2016/**
**\* Williams et al. 2017 http://www.geosci-model-dev.net/10/1291/2017/**

Thanks for the comment, we deleted that part

**Also, the Fader et al. 2015 reference is not the best one for discussing the hydrology-crop linkage in LPJmL, maybe better try**
**\* Rost et al. 2008 http://onlinelibrary.wiley.com/doi/10.1029/2007WR006331/abstract**
**\* Biemans et al. 2011 http://onlinelibrary.wiley.com/doi/10.1029/2009WR008929/abstract**
**\* Jägermeyr et al. 2015 http://www.hydrol-earth-syst-sci.net/19/3073/2015/hess-19-3073-2015.html**

We replaced Feder et al. (2015) with the references that you mentioned.

**and the LPJmL model has been used and tested for regional studies (contrary to your claim on page 6), see**
**\* Biemans et al. 2013 http://www.sciencedirect.com/science/article/pii/S0048969713006451**
**\* Langerwisch et al. 2013 http://www.hydrol-earth-syst-sci.net/17/2247/2013/hess-17-2247-2013.html**

**and it is also no longer true that LPJmL uses country-specific efficiency data, but explicitly simulates the loss terms**

**\* Jägermeyr et al. 2015 http://www.hydrol-earth-syst-sci.net/19/3073/2015/hess-19-3073-2015.html**
**\* Jägermeyr et al. 2016 http://iopscience.iop.org/article/10.1088/1748-9326/11/2/025002/meta**

Thanks for pointing this out, we removed that sentence.

**Lastly, the sentence on page 6 reads as if LPJmL had a simpler implementation of LAI and root development than EPIC, but in fact is uses the EPIC scheme for that.**

Again, thanks for the insightful comment, we removed that statement.

**I'm confident that all of the issues raised by referee #3 and myself can be easily addressed and look forward to the final version of the manuscript.**

**Best regards**
**Christoph Müller**

**Responses to Reviewer # 3**

**This a will written paper describing an interesting coupling of cropsyst and vic. the authors have clearly put in a large effort in revising the paper based on the comment of the reviewers. In general the authors have responded well to the issues raised by the reviewers. There is two issues where to my opinion the response to the review comments could be improved.**

First and foremost, we would like to thank reviewer #3 for her/his helpful comments and suggestions.

**1. Both issue raised the issue that the paper did not discuss other models which also simulated crop and hydrology. In response the author have written a review of the existing models this helps. But I think this also needs to come back in the discussion. Based on the evaluation of the model in which cases in this the best model to use and in which circumstances other model might be better.**

**ADDED/MODIFIED:**

*Model Selection Considerations*

*Which model to apply for a specific research question at hand is dependent on a variety of factors, including geographical considerations but also the level of sophistication needed to address the question. For example, areas with significant*

*irrigation activities can be more precisely simulated with mechanistic irrigation models or areas with cold climate would necessitate models with more sophisticated cold season processes. Also, regional agricultural economic studies require a reliable simulation of crop yield for economically-significant crops grown in the region. Therefore, models that simulate generic C3/C4 crops are not the best option for this type of question. VIC-CropSyst and LPJML are two examples of models*

5    *that can be used to answer this type of question. Moreover, some of the models have been already tested and used for a particular region and resolution, which naturally makes them more reliable for that specific situation.*

**2. The deficit irrigation paragraph is confusing because not it almost seems that deficit irrigation in included in the**

10    **described version of the model. I think it is better to clearly state that the current version of the model and the results presented do not include deficit irrigation and the current version is not connected to a routing module. Maybe in the discussion section this could be addressed and there you could shortly describe how you deficit irrigation can be included.**

We would like to thank the reviewer for this helpful suggestion, we moved the deficient irrigation section to the

15    discussion/application part of the paper and added a sentence that clearly states that deficit scenarios are not included in this manuscript.

**Added/Modified:**

*Simulation of Deficit Irrigation Scenarios*

*Although the results presented in this article do not include results related to deficit irrigation during times of water*

20    *shortage, VIC-CropSyst is able to simulate the impacts of deficit irrigation on hydrologic and cropping systems. VIC-CropSyst's deficit irrigation module requires two main inputs: a) a first approximation to the irrigation water demand obtained by generating time series of irrigation in a zero water stress condition using VIC-CropSyst, and b) deficit fractions that indicate the actual water availability as a function of the crop water requirement. VIC-CropSyst then reads the amount of recorded irrigation from step one and applies the deficit fraction to simulate the agricultural and hydrologic processes*

25    *under realistic water deficit conditions. The deficit fraction can be either homogenously applied across the entire basin or separately specified for each farmer depending on water rights or other considerations. Also, VIC-CropSyst can apply the deficit fraction during different times of the year. For example, if the water deficit happens later in the season, VIC-CropSyst can adjust irrigation amounts according to the timing of water shortage.*

*VIC-CropSyst has also been used in conjunction with reservoir models (e.g., ColSim: Hamlet and Lettenmaier, 1999, and*

30    *YAK-RW: Zagona et al., 2001) to calculate the deficit irrigation fraction (e.g., Barik et al., 2017; Malek, et al., in prep; and Rajagopalan et al., in preparation). In general, the following six steps can be used to calculate and apply a deficit fraction: 1) VIC-CropSyst simulates the hydrologic states such as runoff and base flow as well as the irrigation water demand, 2) a routing model (Lohmann et al., 1998) is used to simulate streamflow, 3) simulated flow is bias-corrected against observed flow, 4) a river system model is used to include operation of dams and reservoir and estimate water availability, 5) the*

*availability of water is compared with water demand, and 6) a deficit fraction is calculated and VIC-CropSyst is run to simulate the impacts of irrigation deficit on the hydrologic cycle and on crop yields.*

**Some other minor issues.**

5    **1. Is the impact of elevated atmospheric CO2 concentration on transpiration included in this model and if yes how?**

CropSyst simulates the impacts of changes in $CO_2$ on crop transpiration, growth and yield; this is done through empirical relationships developed based on results of chamber and open air experiments (Stockle et al., 1992).

**Added/Modified:**

*CropSyst uses empirical relationships (Stockle et al., 1992) to simulate the impact of elevated CO2 concentrations on yield,*

10    *crop growth, and transpiration.*

**2. The figure legends and axis description could be improved.**

**For example. Figure 5 – add to legend that it is total seasonal irrigation, at the Y axis add mm/year**

We thank the reviewer for pointing out these issues; we have corrected them in the manuscript.

15    **Added/Modified:**

*Figure 5-Simulated versus recorded total seasonal irrigation water in an irrigated corn field at the NE flux tower site.*

[Figure]

20    **In figure 6 – I think it is average monthly evaporation, the unit is probable mm/day add this to Y axis legend**

Thanks for the comment we corrected the Y-axis label and mentioned it in the legend as well

**Added/Modified:**

[Figure]

*Figure 6- Comparison of simulated and observed corn evapotranspiration (ET; mm/day) at two flux tower sites located in NE and IL. The NE site is irrigated while IL is a non-irrigated field.*

**Figure 7 use metric values not tons!**

We have corrected this issue.

[Figure]

10 **In general white backgrounds work the best for figures.**

We changed the backgrounds to white.

[revised manuscript text omitted]

The following table appears within the figure:

|  | VIC-offline | VIC-CropSyst | Change in Error |
|---|---|---|---|
| Error Over Irrigated Cells | 28.4% | 17% | 11.40% |

**Figure 10 – Comparison of simulated and empirically-derived ET over the U.S. Pacific Northwest. The simulation and observation period is 1982-2008. Panel a shows observed ET (Liu et al., 2013), panels b and c show the simulated ET using VIC-offline and VIC-CropSyst, respectively. Panel d shows where the irrigated areas are located in these three states. Panels e to h show relative and absolute errors of simulated ET by VIC-offline and VIC-CropSyst.**

[Figure]

**Figure 11- Regional application of VIC-CropSyst in conjunction with a river system model (YAK-RW; Hubble, 2012; Zagona et al., 2001) and an economic model to simulate historical (1981-2006) drought frequency (panel a), when the percentage of the water right allocated for the irrigation season (i.e., proration rate) is lower than 70%, 80% and 90%. Panels b and c (Malek et al., in prep) show the percentage of farmers (perennial crop growers) who invest in new efficient irrigation technologies in response to simulated droughts during the two decades of 1990-2000 (panel b) and 2050-2060 (panel c).**